# LOOKING LOCALLY: OBJECT-CENTRIC VISION TRANSFORMERS AS FOUNDATION MODELS FOR EFFICIENT SEGMENTATION

## ABSTRACT

Current state-of-the-art segmentation models encode entire images before focusing on specific objects. As a result, they waste computational resources—particularly when small objects are to be segmented in high-resolution scenes. We introduce FLIP (Fovea-Like Input Patching), a parameter-efficient vision model that realizes object segmentation through biologically-inspired top-down attention. FLIP selectively samples multi-resolution patches centered on objects of interest from the input. As a result, it allocates high-resolution processing to object centers while maintaining coarser peripheral context. This off-grid, scale-invariant design enables FLIP to outperform META's Segment Anything models (SAM, SAM2 and fast variants) by large margins: With more than $440\times$ fewer parameters, FLIP-Tiny (0.51M parameters) reaches a mean IoU of 79.90% while SAM2-L reaches 75.87% IoU (224.45M parameters). FLIP-Large even achieves 83.26% mean IoU (96.6M parameters), still running about $2\times$ faster than SAM2-L. We evaluate on six benchmarks in total. In five established benchmarks (Hypersim, KITTI-360, OpenImages, COCO, LVIS) FLIP consistently outperforms SAM and various variants of it. In our novel ObjaScale dataset, which stress-tests scale invariance with objects ranging from 0.0001% up to 25% of the image area, we show that FLIP segments even very small objects accurately, where existing models fail severely. FLIP opens new possibilities for real-time, object-centric vision applications and offers much higher energy efficiency. We believe that FLIP can act as a powerful foundation model, as it is very well-suited to track objects over time, for example, when being integrated into slot-based scene segmentation architectures.

## 1 INTRODUCTION

Object-centric models have emerged as a powerful paradigm for structured perception in visual tasks. They offer the potential to represent complex scenes in a more interpretable and compositional manner. While traditional architectures such as Convolutional Neural Networks (CNNs) (Liu et al., 2022) and Vision Transformers (ViTs) (Dosovitskiy et al., 2020) have demonstrated impressive performance on large-scale datasets, they often lack the nuanced object-level understanding required for robust scene parsing. Furthermore, these models typically require massive amounts of labeled data and exhibit vulnerabilities to adversarial perturbations.

Recent advances in object-centric learning, such as Slot Attention (Locatello et al., 2020), have sought to address these challenges by enabling the model to discover and represent objects within a scene as distinct entities. Models like SAVi++ (Elsayed et al., 2022), VideoSAUR (Zadaianchuk et al., 2024), and Loci (Traub et al., 2023; 2024a;b) have advanced the state-of-the-art in unsupervised object-centric learning. They are able to disentangle objects from complex backgrounds and track their identities over time. Still, these approaches struggle to scale effectively to more complex, real-world data.

In contrast, the Segment Anything (SAM) model and its successor SAM2 (Kirillov et al., 2023; Ravi et al., 2024) have introduced a paradigm shift in object-centric learning. SAM learns from a vast array of diverse data that is segmented by a two stage segmentation process. First, a powerful transformer-based foundational model encodes the complete image. Second, a query-based focusing

mechanism specifies which object to segment. Only this second mechanism targets one image area or object and leads to the production of the targeted output mask. SAM2 marks the state-of-the-art in object segmentation tasks. However, despite its impressive performance, SAM models have their limits. First, the transformer-based encoder requires very large computational resources. Second, the encoder backbone encodes the complete image, potentially wasting processing resources, particularly when small objects are to be segmented.

Derivative model variants like EfficientSAM, MobileSAM, and FastSAM (Xiong et al., 2023; Zhang et al., 2023; Zhao et al., 2023) address the former limitation. In our evaluation of single-object segmentation performance across multiple datasets, EfficientSAM employs SAMI (SAM-leveraged masked image pretraining), training lightweight ViT encoders to reconstruct features from SAM's ViT-H encoder, and achieves 72.29% mean IoU with only 10.22M parameters (EfficientSAM-T) compared to SAM2-L's 75.87% IoU with 224.45M parameters. MobileSAM applies decoupled distillation, replacing SAM's ViT-H encoder (632M parameters) with TinyViT (5M parameters) (Wu et al., 2022), achieving 71.33% mean IoU with an over $60\times$ parameter reduction and $\sim$21ms runtime per image. FastSAM replaces the first-stage transformer-based encoder in SAM with a convolutional ANN (CNN) that is pre-trained to segment full images. It achieves a runtime per image of under 10ms but exhibits significantly reduced performance with only 44.58% mean IoU (FastSAM-s).

At this point, nearly all segmentation techniques, including SAM, SAM2 and its efficient variants, rely on a full image encoder. This is also the case for most object-centric models, such as Slot Attention, VideoSAUR, and related work (Locatello et al., 2020; Elsayed et al., 2022; Singh et al., 2022; Zadaianchuk et al., 2024), which first encode the entire image before assigning information to slots. The challenge to computationally efficiently segment and track small but potentially high-resolution objects across diverse and complex scenes remains. Here, we introduce a vision transformer-based foundational model that utilizes a local, pixel-patch-based image encoder to generate accurate masks.

In particular, we introduce FLIP: a fovea-like input patching approach that is integrated in an object-centric, off-grid vision framework. FLIP dynamically adapts its processing pipeline to the object's size and spatial characteristics. It ignores currently irrelevant image subregions and focuses on critical regions with a flexible, multi-resolution approach. Our key contributions are:

- **Off-Grid, Scale-Invariant Object Encoding**: We introduce a fovea-inspired pixel-based patch sampling method that directly encodes image regions off-grid, adaptively focusing on objects of interest in a multi-resolution fashion. This scale-invariant approach is robust to large variations in object size and image resolution. It enables the detailed encoding of very small objects even in high-resolution scenes.

- **Exceptional Parameter and Computational Efficiency**: FLIP achieves superior segmentation performance while using orders of magnitude fewer parameters than existing state-of-the-art models. With lightweight encoders and efficient processing, FLIP provides significant energy savings and faster inference times, making it suitable for real-time applications and resource-constrained environments.

- **State-of-the-Art Segmentation Performance with High Parameter Efficiency**: Despite using significantly fewer parameters compared to state-of-the-art models from the SAM family, FLIP achieves superior segmentation accuracy on standard benchmarks such as Hypersim, KITTI-360, OpenImages, COCO and LVIS.

- **Superior performance on novel dataset *ObjaScale*** We introduce *ObjaScale*, a high-resolution evaluation benchmark with Blender-rendered objects at varying scales on real-world High Dynamic Range Image (HDRI) backgrounds, designed to stress-test scale invariance. As in the other datasets, FLIP outperforms all SAM variants on ObjaScale and shows that it can segment even rather tiny objects accurately.

## 2  RELATED WORK

Several lines of research contain methods that share thematically similar ideas in handling multi-scale objects, dynamic sampling, or biologically inspired foveation. *Deformable Convolutional Networks* (Dai et al., 2017) and their successors Zhu et al. (2019); Xiong et al. (2024), as well as deformable-attention vision transformers such as DAT (Xia et al., 2022), introduce learnable offsets or selective kernels to better handle varying spatial structures. *Focal Sparse Convolutional Networks* (Chen

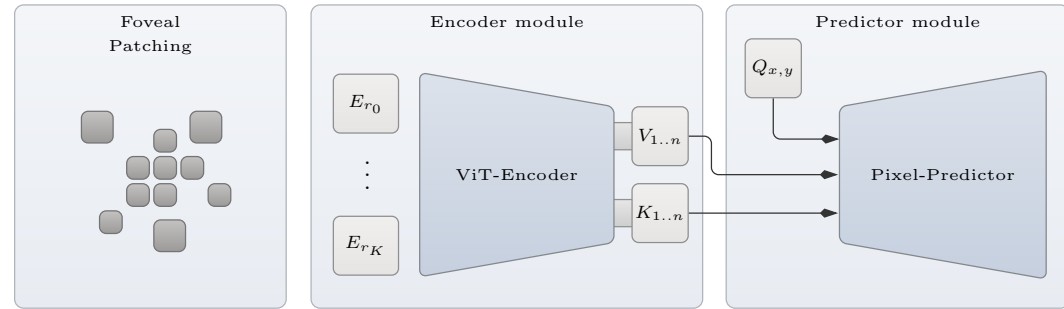

Figure 1: Overview of the FLIP architecture. The Foveal Patching module dynamically samples multi-resolution patches centered around objects of interest. These patches are embedded into a unified latent space using resolution-specific Patch Embedding Modules $(E_{r_0}, \ldots, E_{r_K})$. The Vision Transformer Encoder processes the embedded patches, generating keys $K_{1..n}$ and values $V_{1..n}$. The Pixel-Predictor performs attention over queries derived from pixel coordinates $Q_{x,y}$, enabling instance segmentation with pixel-level precision.

et al., 2022) focus computation on salient 3D regions in point clouds. Additionally, biologically inspired foveation and overview-first / look-closely-next processing have been explored in works like Lukanov et al. (2021); Kaplanyan et al. (2019); Thavamani et al. (2021) and in recent backbones such as OverLoCK (Lou & Yu, 2025) and TransNeXt (Shi, 2024). Most closely related to our setting, Segment This Thing (STT) (Schmidt & Newcombe, 2025) applies a fixed, point-centered foveated tokenization to reduce encoder tokens, whereas FLIP performs object-conditioned, off-grid multi-resolution patch sampling driven by a 2D Gaussian prior.

## 3  METHODS

In this section, we present the overall Fovea-Like Input Patching (FLIP) architecture. Its main processing pipeline is shown in Figure 1. FLIP is a supervised vision model for efficient object-centric segmentation. It combines a fovea-inspired patching mechanism with a Vision Transformer (ViT)-based encoder. The encoded information is then used to generate the targeted object's segmentation mask.

The fovea-like patching mechanism dynamically selects multi-resolution patches based on a 2D Gaussian distribution, which has an effect similar to the query in SAM variants but acts directly on the input image. High-resolution patches focus on the object center, capturing fine details, while coarser patches cover peripheral regions, which inform FLIP about the surrounding context.

The ViT encoder processes the sampled patches and outputs latent tokens from which we compute keys $K_{1..n}$ and values $V_{1..n}$. The Pixel-Predictor then computes queries $Q_{x,y}$ from pixel coordinates and uses an attention mechanism to predict the segmentation mask with pixel level accuracy (see 8c).

FLIP is trained end-to-end with a sparse supervised loss for mask prediction. In the following subsections, we detail the fovea-like sampling mechanism and summarize the ViT encoder and Pixel-Predictor; full mathematical descriptions of the latter are provided in Appendix A.1.

### 3.1  FOVEA-LIKE INPUT PATCHING

We equip our ViT-based model with a multi-resolution, fovea-inspired patching mechanism that centers around the object of interest. The mechanism preserves high-resolution detail at the object center and coarser coverage in peripheral regions. Specifically, we derive the object center $\mu = (\mu_x, \mu_y)$ and covariance $\Sigma$ from the ground-truth mask, yielding a 2D Gaussian $\mathcal{N}(\mu, \Sigma)$ that approximates the object's spatial extent and orientation in compressed form. For input sampling, the Gaussian serves as a spatial input query—similar to the prompts in SAM variants—from which we then draw patches at multiple scales.

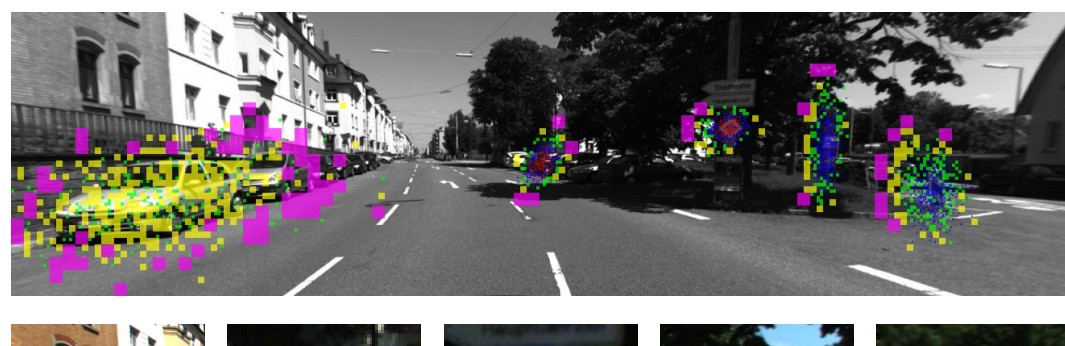

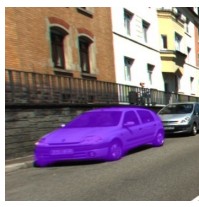 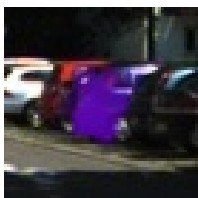 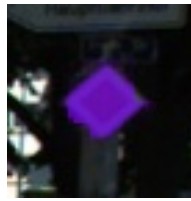 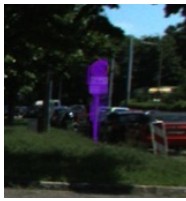 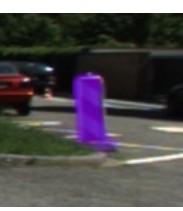

Figure 2: Visualization of our FLIP (Fovea-Like Input Patching) approach applied to an image from the KITTI-360 dataset, showcasing potential applications in autonomous driving. The figure illustrates how our model dynamically focuses on multiple objects within a complex urban scene by allocating multi-resolution patches centered around estimated object locations. Higher-resolution patches (smaller sizes) are concentrated on critical areas such as vehicles and road signs, emulating a foveal vision system, while lower-resolution patches (larger sizes) cover peripheral regions to enable the consideration of the surrounding context. Patches are color-coded by size: purple for $16 \times 16$ patches, yellow for $8 \times 8$, green for $4 \times 4$, blue for $2 \times 2$, and red for $1 \times 1$.

We define $K$ patch sizes $p_1 < p_2 < \cdots < p_K$, from smallest (highest resolution) to largest. We fix the total number of patches $N$ and choose $N_i$ patches for patch size $p_i$ as follows. First, we compute default numbers $\hat{N}_i$ by approximating the integral of the 2D Gaussian and dividing by $p_i^2$, scaled by a coverage parameter $c \in (0.1, 2)$ that controls the density of patch sampling. Then, starting at the coarsest resolution (i.e., $p_K$), we compute $N_i = \min(\hat{N}_i, N - \sum_{j=i+1}^{K} N_j)$. As a result, the chosen patches cover the Gaussian inner area with a density proportional to $c$ and the number of chosen patches per size distributes itself from coarse to fine until the total number of patches $N$ is reached ($N = \sum_i^K N_i$).

We sample $N_i$ patches from $\mathcal{N}(\mu, \Sigma)$ with continuous coordinates, allowing patches to be centered at real-valued positions $(x, y) \in \mathbb{R}^2$. Pixel values are extracted using bilinear interpolation, enabling sub-pixel precision. To control patch overlap, we employ a spatial hash grid that tracks placed patches. When sampling, we reject candidate positions that would result in overlap exceeding the maximum overlap threshold $\tau \in [0, 4]$. This ensures efficient spatial coverage while allowing for controlled redundancy. We implement this sampling mechanism as a Python C extension in the data loading pipeline, achieving approximately 1ms average extraction time per image. Each sampled patch is first flattened and then mapped to a common embedding space via resolution-specific encoders $E_{r_i}$ (similarly to Jaegle et al. (2021)), yielding a set of tokens $T = \{t_1, ..., t_N\}$ of equal tensor size. These embedded tokens are concatenated and fed to the main ViT layers. Note that the approach is in principle independent of the full image size, because sampling depends on $\mu$ and $\Sigma$ only.

### 3.1.1 PATCH SAMPLING DETAILS

Algorithm 1 summarizes the extraction routine: $\mu$ and $\Sigma$ parametrize the Gaussian object query, $p_1 < \cdots < p_K$ denotes the $K$ patch sizes, $N$ the total patch budget, $c$ the coverage factor, and $\tau$ the maximal allowed overlap count per patch.

The helper INITSPATIALHASH instantiates the bounding-box hash grid such that every accepted patch overlaps previous ones at most $\tau$ times. BILINEARCROP evaluates the RGB values on a $p_i \times p_i$ grid centered at $(x, y)$, enabling sub-pixel sampling of the Gaussian draw; $\tilde{\mathbf{x}}$ stores the normalized center coordinates used to build the positional encodings, i.e., $\tilde{x} = (x - W/2)/128$ and $\tilde{y} = (y - H/2)/128$.

---

**Algorithm 1** Fovea-like patch sampling

---

**Require:** Image $I \in [0, 255]^{H \times W \times C}$, Gaussian query $(\mu, \Sigma)$, patch sizes $\{p_i\}_{i=1}^K$, patch budget $N$,
   coverage $c \in [0.1, 2]$, overlap threshold $\tau \in [0, 4]$
**Ensure:** Per-resolution patch buffers $\{P_i\}_{i=1}^K$, normalized coordinates $\tilde{\mathcal{X}}$
1: $(\sigma_x, \sigma_y, R) \leftarrow \text{eig}(\Sigma)$                                     ▷ principal axes and rotation
2: area $\leftarrow 2\pi\sigma_x\sigma_y$                                                      ▷ integral of $\mathcal{N}(\mu, \Sigma)$
3: **for** $i = 1, \ldots, K$ **do**
4:     $\hat{N}_i \leftarrow \lfloor c \cdot \text{area}/p_i^2 \rfloor$
5: **end for**
6: **for** $i = K, \ldots, 1$ **do**
7:     $N_i \leftarrow \min\left(\hat{N}_i, N - \sum_{j=i+1}^K N_j\right)$                      ▷ coarse-to-fine allocation
8: **end for**
9: $\mathcal{H} \leftarrow \text{INITSPATIALHASH}(\{p_i\}, \{N_i\}, \tau)$
10: $t \leftarrow 0$                                                                           ▷ cumulative target index
11: **for** $i = 1, \ldots, K$ **do**                          ▷ $p_1$ highest resolution; $|P_i|$ counts accepted patches
12:     attempts $\leftarrow 0$
13:     **while** $|P_i| < N_i$ **and** attempts $< 10N_i$ **do**
14:         attempts $\leftarrow$ attempts $+ 1$
15:         $(x, y) \sim \mathcal{N}(\mu, \Sigma)$                                             ▷ sample continuous patch center
16:         **if** $(x, y, p_i)$ not entirely within $I$ **then**
17:             **continue**
18:         **end if**
19:         **if** $\text{OVERLAP}(\mathcal{H}, (x, y, p_i)) > \tau$ **then**
20:             **continue**
21:         **end if**
22:         $\mathcal{H} \leftarrow \text{ADDBOX}(\mathcal{H}, (x, y, p_i))$
23:         $P \leftarrow \text{BILINEARCROP}(I, (x, y), p_i)$
24:         $\tilde{\mathbf{x}} \leftarrow \left((x - W/2)/128, (y - H/2)/128\right)$           ▷ normalized patch-coordinates
25:         Append $(P, \tilde{\mathbf{x}}, t + |P_i|)$ to buffers at resolution $i$
26:     **end while**
27:     $t \leftarrow t + |P_i|$
28: **end for**

---

If the rejection steps exhaust the $10N_i$ attempts before filling $N_i$, the collected count $|P_i|$ is kept. Finally, every patch tensor is flattened and fed through its resolution-specific encoder $E_{r_i}$, producing $T = \{t_1, \ldots, t_{\sum_i |P_i|}\}$, the token sequence consumed by the transformer-based encoder.

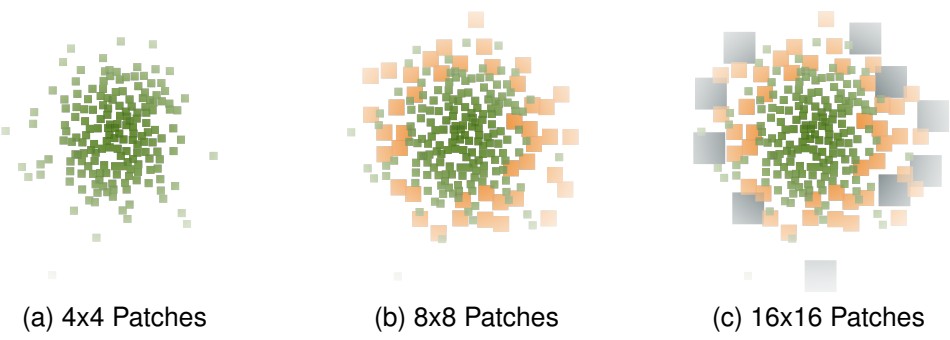

(a) 4x4 Patches          (b) 8x8 Patches          (c) 16x16 Patches

Figure 3: Example visualization of the per-resolution iterative sampling process: First 4×4 patches are sampled, next 8×8 patches, and last 16×16 patches. Since smaller-resolution patches are sampled first and we control for a specific maximum overlap, larger patches naturally are sampled near the perimeter, and thus the fovea-like patching emerges.

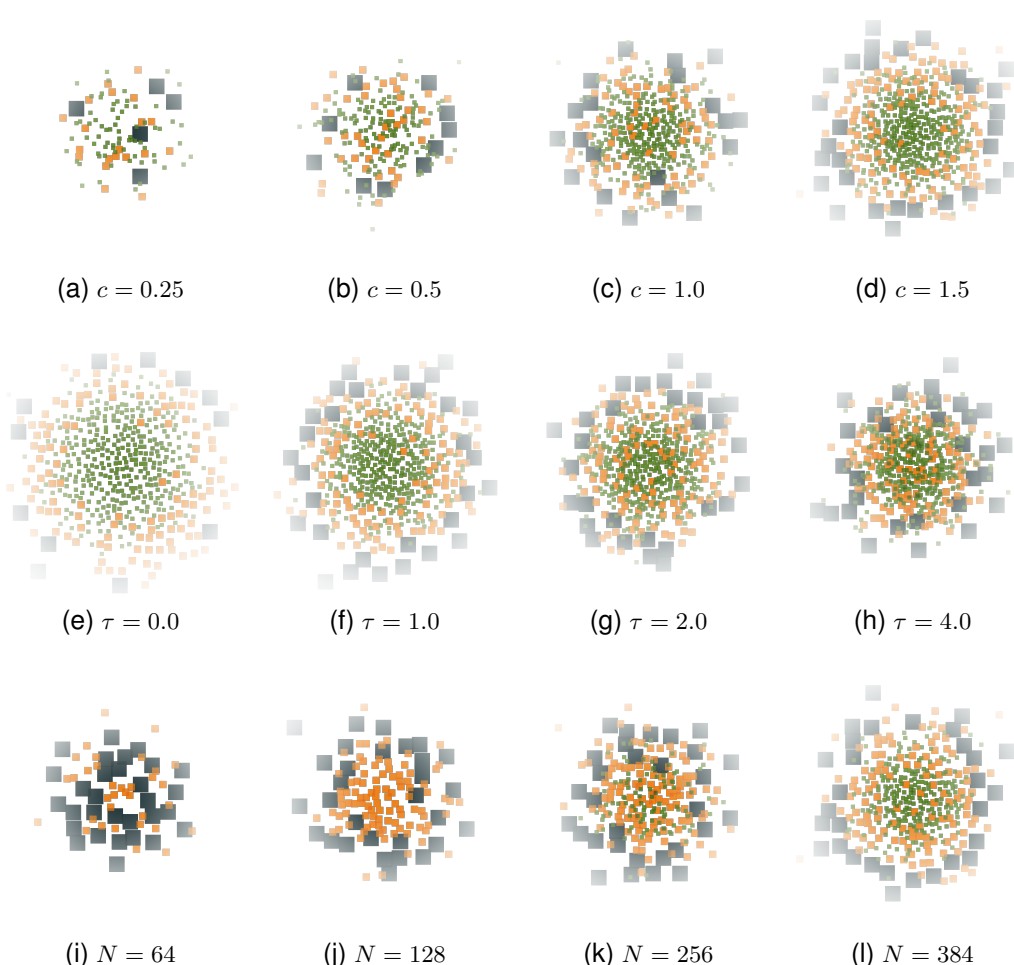

Figure 4: Example of sampled patches from an isotropic Gaussian with varying coverage $c$, the maximal allowed overlap count per patch $\tau$ and patch budget $N$. **a-d** varying coverage $c$ with fixed $\tau = 1.5$ and $N = 512$. The coverage also affects the number of sampled patches and leads to an overall sparser patching of the input for $c < 1$. **e-h** varying $\tau$ with fixed $c = 1.5$ and $N = 512$. Allowing for more overlap between patches leads to a denser sampling of the input. **i-l** varying $N$ with fixed $c = 1.5$ and $\tau = 1.5$ directly affects the information density of the input sampling with more patches leads to a more fined grained input.

### 3.2 ENCODER AND PIXEL-LEVEL MASK PREDICTION

The sampled patch tokens are processed by a ViT-based encoder with LaPE-style positional embedding injection and separate layer normalizations for queries, keys, and values. Encoder outputs feed into a pixel-level predictor that builds positional queries from target coordinates and attends to the encoded patch tokens to produce mask logits. Detailed mathematical formulations and architectural schematics, including attention block and pixel predictor diagrams, are provided in subsection A.1 ( 8b, 8c).

### 3.3 TRAINING

We train FLIP exclusively on META's SA-1B Dataset (Kirillov et al., 2023) with four model variants: FLIP-Tiny (0.51M parameters), FLIP-Small (2.3M parameters), FLIP-Middle (11.5M parameters), and FLIP-Large (96.6M parameters), performing approximately 8.5M, 7.4M, 5.8M and 2.7M updates

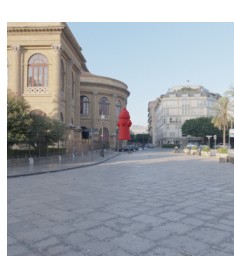 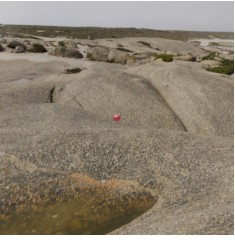 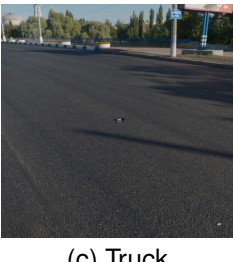 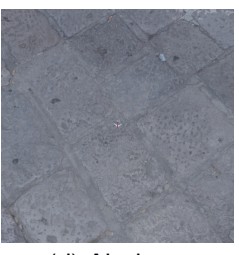

(a) Fire Hydrant    (b) Apple    (c) Truck    (d) Airplane

Figure 5: Examples from our synthetic dataset. Objects from various categories are rendered with high-resolution HDRI Haven backgrounds.

respectively with a batch size of 256. These models are significantly more parameter-efficient than SAM variants (up to 641.1M parameters).

To fairly compare against bounding-box-prompted baselines, we also report FLIP$_{bb}$-* variants that consume 2D Gaussian prompts constructed from bounding boxes rather than mask-derived Gaussians. The box midpoint defines $(\mu_x, \mu_y)$, $\sigma_x$ and $\sigma_y$ are set to fixed fractions of the box width and height (0.188 and 0.34) and the covariance stays axis-aligned. Each FLIP$_{bb}$ model is fine-tuned on SA-1B with these box-derived Gaussians to match the information content available to the other bounding-box baselines.

For data augmentation, we apply random perturbations to the 2D Gaussian derived from ground truth masks by shifting center coordinates, stretching or compressing $\sigma_x$ and $\sigma_y$, and introducing small rotations. The number of sampled tokens $N$ is varied around $\mu = 512$ during training to improve generalization. Additionally the coverage $c$ parameter is randomly sampled from $[0.1, 2]$ and the overlap threshold parameter $\tau$ is sampled from $[0, 4]$.

In a final learning phase we finetune FLIP variants either with bounding box derived prompts or full 2d Gaussian prompts without data augmentation. In this phase, we fix $N = 512$, $c = 1.44$ and $\tau = 2.67$. The coverage and overlap values were chosen to maintain a high information density during the patch sampling process.

**Sparse Mask Prediction.** Instead of dense mask prediction, we employ a computationally efficient sparse sampling strategy that focuses on boundary regions. Pixels are grouped into seven distance-based categories according to their L1 distance from mask boundaries: [0-2), [2-4), [4-8), [8-16), [16-32), [32-64), and [64-∞) pixels. For each sample, we select 2048 pixels distributed across these groups with equal sampling from inside and outside the mask to maintain class balance.

We implement adaptive sampling that dynamically redistributes pixels across distance groups based on prediction difficulty. Groups with lower IoU performance receive more samples in subsequent iterations, ensuring computational focus on challenging boundary regions.

The sparse mask loss reduces computational complexity from $O(HW)$ to $O(k)$ where $k \ll HW$:

$$\mathcal{L} = \text{BCE}(\hat{M}_{\text{sparse}}, M_{\text{sparse}}) \tag{1}$$

where $\hat{M}_{\text{sparse}}$ and $M_{\text{sparse}}$ represent predicted and ground truth values at sampled locations.

### 3.4 INFERENCE

During inference, FLIP employs two strategies to efficiently generate segmentation masks while maintaining the sparse computation paradigm.

**5-Sigma Bounding Box Prediction** Instead of predicting mask values for all pixels in the image, we exploit the Gaussian prior to restrict predictions to a 5-sigma bounding box around the object center. Given the Gaussian parameters $(\mu_x, \mu_y, \sigma_x, \sigma_y)$ from the input prompt, we compute a bounding box using $\sigma_{\text{iso}} = \max(\sigma_x, \sigma_y)$ and sample pixels only within the region $[\mu_x \pm 5\sigma_{\text{iso}}, \mu_y \pm 5\sigma_{\text{iso}}]$. This

Table 1: Comparison of Mean IoU (%) and Std IoU (%) across different datasets.

| Model | Size (M) | Time (ms) | Hypersim | KITTI-360 | OpenImage | COCO | LVIS | ObjaScale (ours) | Average |
|---|---|---|---|---|---|---|---|---|---|
| | | | Mean ± Std IoU (%) | Mean ± Std IoU (%) | Mean ± Std IoU (%) | Mean ± Std IoU (%) | Mean ± Std IoU (%) | Mean ± Std IoU (%) | Mean ± Std IoU (%) |
| FastSAM-s | 11.8 | 9.94 | 36.80 ± 34.62 | 37.71 ± 32.76 | 61.14 ± 32.10 | 46.51 ± 33.73 | 37.11 ± 35.19 | 48.20 ± 35.69 | 44.58 ± 34.02 |
| FastSAM-x | 72.2 | 24.32 | 34.36 ± 34.90 | 38.92 ± 34.06 | 69.31 ± 29.25 | 55.31 ± 34.55 | 43.24 ± 37.47 | 47.11 ± 36.41 | 48.04 ± 34.44 |
| MobileSAM | 10.13 | 21.15 | 68.03 ± 21.36 | 64.64 ± 18.87 | 82.20 ± 16.74 | 73.80 ± 17.48 | 71.59 ± 21.36 | 67.72 ± 25.03 | 71.33 ± 20.14 |
| EfficientSAM-T | 10.22 | 26.75 | 68.35 ± 24.52 | 60.82 ± 24.72 | 84.32 ± 15.08 | 75.93 ± 17.04 | 75.56 ± 19.95 | 68.74 ± 26.43 | 72.29 ± 21.29 |
| EfficientSAM-S | 26.41 | 47.98 | 69.65 ± 22.20 | 64.88 ± 19.35 | 85.99 ± 14.54 | 77.07 ± 16.81 | 75.18 ± 21.06 | 67.80 ± 26.76 | 73.43 ± 20.12 |
| SAM-B | 93.7 | 72.67 | 71.46 ± 20.88 | 62.38 ± 21.41 | 84.72 ± 15.38 | 76.07 ± 16.95 | 76.93 ± 19.17 | 71.38 ± 25.36 | 73.82 ± 19.86 |
| SAM-L | 312.3 | 148.78 | 72.13 ± 21.21 | 62.73 ± 20.31 | 86.94 ± 13.41 | 78.19 ± 16.05 | 77.93 ± 19.43 | 72.68 ± 25.22 | 75.10 ± 19.27 |
| SAM-H | 641.1 | 232.04 | 72.37 ± 21.65 | 62.47 ± 20.52 | 87.06 ± 13.53 | 78.41 ± 16.15 | 78.36 ± 19.50 | 73.76 ± 24.59 | 75.41 ± 19.32 |
| SAM2-T | 38.96 | 30.38 | 71.56 ± 20.50 | 66.80 ± 20.40 | 87.08 ± 12.79 | 78.55 ± 15.37 | 76.85 ± 19.31 | 69.89 ± 26.28 | 75.12 ± 19.11 |
| SAM2-S | 46.06 | 33.43 | 71.37 ± 21.33 | 66.43 ± 21.83 | 87.61 ± 12.66 | 78.82 ± 15.54 | 77.16 ± 19.40 | 69.68 ± 26.43 | 75.18 ± 19.53 |
| SAM2-B+ | 80.85 | 46.43 | 72.07 ± 20.53 | 65.77 ± 19.32 | 87.96 ± 11.97 | 79.35 ± 15.11 | 77.33 ± 19.46 | 70.50 ± 26.53 | 75.50 ± 18.82 |
| SAM2-L | 224.45 | 88.64 | 72.89 ± 20.53 | 67.30 ± 20.78 | 88.64 ± 11.35 | 79.63 ± 15.01 | 77.20 ± 19.66 | 69.54 ± 27.00 | 75.87 ± 19.05 |
| FLIP-Tiny | **0.51** | 9.68 | 74.14 ± 21.03 | 72.62 ± 16.56 | 86.74 ± 12.59 | 79.83 ± 14.27 | 77.55 ± 20.02 | 88.55 ± 15.08 | 79.90 ± 16.59 |
| FLIP-Small | 2.3 | 11.94 | 75.59 ± 20.95 | **72.98** ± 16.83 | 88.96 ± 10.56 | 81.32 ± 14.02 | 79.12 ± 20.27 | 89.67 ± 14.91 | 81.27 ± 16.26 |
| FLIP-Middle | 11.5 | 18.00 | 79.14 ± 19.69 | 72.21 ± 16.90 | 90.84 ± 8.80 | **82.86** ± 12.89 | 82.61 ± 17.78 | **90.70** ± 13.86 | 83.06 ± 14.99 |
| FLIP-Large | 96.6 | 44.13 | **80.34** ± 19.28 | 70.72 ± 17.51 | **91.36** ± 7.72 | 82.66 ± 13.85 | **83.83** ± 16.94 | 90.65 ± 13.80 | **83.26** ± 14.85 |
| FLIP$_{bb}$-Tiny | **0.51** | **8.81** | 68.32 ± 21.37 | 66.77 ± 17.92 | 83.08 ± 14.58 | 75.38 ± 16.03 | 75.50 ± 19.72 | 86.80 ± 16.41 | 75.97 ± 17.67 |
| FLIP$_{bb}$-Small | 2.3 | 10.63 | 70.53 ± 21.15 | 67.09 ± 18.07 | 85.76 ± 12.86 | 77.48 ± 15.49 | 77.93 ± 18.96 | 88.39 ± 15.99 | 77.86 ± 17.09 |
| FLIP$_{bb}$-Middle | 11.5 | 14.28 | 72.21 ± 20.87 | 67.53 ± 18.34 | 87.67 ± 11.46 | 78.77 ± 15.77 | 79.90 ± 18.18 | 88.93 ± 15.63 | 79.17 ± 16.71 |
| FLIP$_{bb}$-Large | 96.6 | 31.45 | 72.83 ± 20.83 | 66.93 ± 19.14 | 88.59 ± 10.64 | 78.69 ± 16.56 | 80.79 ± 17.69 | 88.91 ± 15.65 | 79.46 ± 16.75 |

reduces the number of mask queries from $O(HW)$ to $O(k^2)$ where $k \propto \sigma_{\text{iso}}$, providing significant speedup for small objects while maintaining full accuracy for larger ones.

**Bounding-Box Derived 2D Gaussian Prompts** When an axis-aligned bounding box is provided as the input prompt, we convert it into a 2D Gaussian prompt (center at the box midpoint, $\sigma_x$ and $\sigma_y$ set as fixed fractions of the box width and height, covariance kept axis-aligned). We then predict mask values only for pixels inside the prompted 2D bounding box. Pixels outside the box are never queried and their values in the predicted final mask are set to zero.

## 4 RESULTS

We evaluate FLIP on five standard benchmarks: Hypersim, KITTI-360, OpenImages, COCO and LVIS (Roberts et al., 2021; Liao et al., 2022; Kuznetsova et al., 2020; Lin et al., 2014; Gupta et al., 2019). Additionally, we evaluate FLIP on our newly constructed synthetic dataset ObjaScale, designed to stress-test scale invariance. Like SAM, FLIP was trained on the SA-1B dataset (Kirillov et al., 2023), but was neither trained nor fine-tuned on the ObjaScale or any of the other evaluation datasets. Our experiments compare FLIP against state-of-the-art segmentation methods namely SAM2 (Ravi et al., 2024), SAM (Kirillov et al., 2023), EfficientSAM (Xiong et al., 2023), MobileSAM (Zhang et al., 2023), and FastSAM (Zhao et al., 2023). Additionally, we compare our method to another foveated segmentation approach, STT (Schmidt & Newcombe, 2025). However, this comparison is not directly comparable to the other segmentation baselines, since STT relies solely on point prompts, which are ambiguous regarding whether they indicate an entire object or only a part of it. In contrast, bounding-box or 2D-Gaussian prompts are unambiguous in this respect. Therefore we only report STT evaluation results on ObjaScale in Figure 6, where the identification of the object in question is much easier in comparison to the other datasets and the challenge is the extreme scale invariance. Besides comparing the overall performance, we focus on handling small objects, achieving high IoU, and balancing parameter/runtime trade-offs.

### 4.1 EXPERIMENTAL SETUP AND DATASET CREATION

We selected 68 diverse categories from Objaverse (Deitke et al., 2023) and combined each with high-resolution HDRI Haven (Zaal, Greg) backgrounds in Blender, yielding 10,200 synthetic images, which form our ObjaScale dataset. Image resolution was randomized from 512 to 8192 pixels, while object dimensions were constrained to a maximum width/height of 256 pixels, producing masks spanning minuscule ($< 0.0001\%$) to large fractions of the image area (up to 25%) (Figure 5). For SAM variants, bounding-box prompts computed from ground-truth masks were used while FLIP employed 2D Gaussian prompts by design. For STT, we follow the point-prompt selection procedure described in their paper: the prompt is chosen as the point inside the ground-truth mask that lies farthest away from any mask boundary.

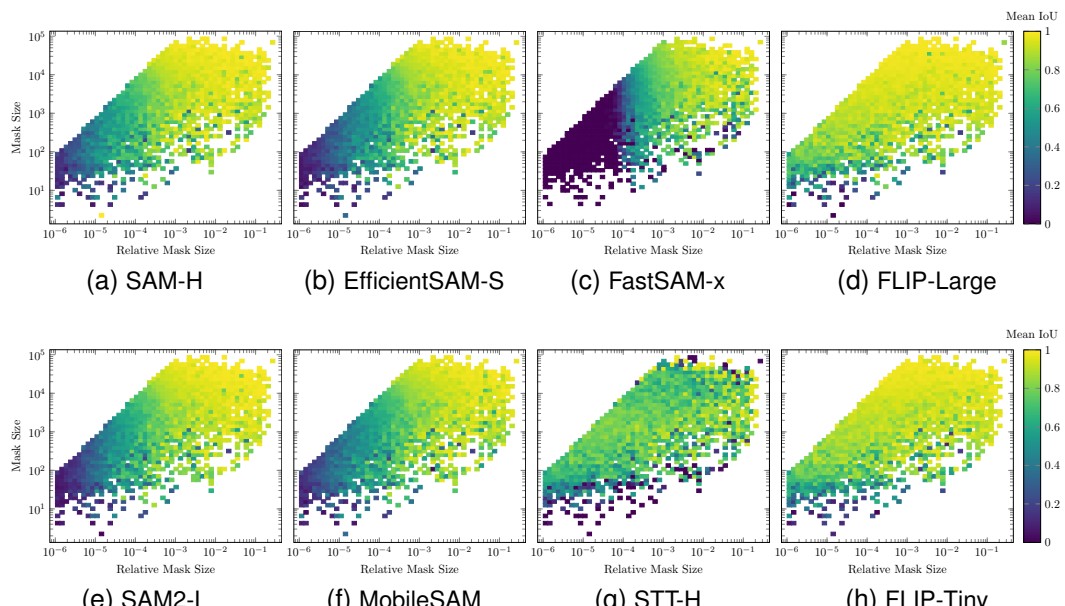

Figure 6: Heatmaps illustrating IoU performance in relation to relative (X-axis) vs. absolute mask size (Y-axis) on the ObjaScale dataset. The SAM-variants' full-image methods lose performance the smaller the object is in relation to the overall image size (left side of the graphs), with FastSAM-x failing completely for objects smaller than 0.1% of the image area. Only foveated methods like STT and FLIP maintain performance for relatively small objects. Notably, FLIP still maintains performance for relatively large objects in comparison with STT and is also stronger when the absolute mask sizes reach the detection limit at masks with fewer than 100 pixels (bottom part of the graphs).

## 4.2 OVERALL SEGMENTATION PERFORMANCE

We denote bounding-box–prompted variants with the subscript "$_{bb}$" (e.g. FLIP$_{bb}$-Small); these models consume 2D Gaussian prompts derived from bounding boxes to match the information available to box-prompted baselines.

Table 1 presents results across all datasets. FLIP consistently outperforms SAM variants with fewer parameters. FLIP-Large attains 83.26% mean IoU with 96.6M parameters, outperforming SAM-H's 75.41% IoU (641.1M parameters) and SAM2-L's 75.87% IoU (224.5M parameters)—a $6.6\times$ and $2.3\times$ parameter reduction with improved accuracy. FLIP-Tiny (0.51M parameters) achieves 79.90% mean IoU, surpassing all SAM variants including SAM2-L.

FLIP's advantage is particularly pronounced on ObjaScale, achieving 88.6–90.7% IoU versus SAM-H's 73.76%, demonstrating superior scale invariance across diverse object sizes. Notably, SAM2 dropped in performance on ObjaScale in comparison to SAM. Only STT, the other foveated segmentation model we evaluated, can maintain consistent performance on ObjaScale even for relatively small objects (see Figure 6), but it is still over 15 percentage points behind FLIP. On established benchmarks, FLIP maintains competitive performance with 91.36% IoU on OpenImages (vs. SAM2-L's 88.64%) and 82.66% on COCO (vs. SAM2-L's 79.63%).

## 4.3 PARAMETER EFFICIENCY AND PERFORMANCE

Figure 7a illustrates the parameter efficiency of FLIP variants. While maintaining superior segmentation quality, FLIP models operate with orders of magnitude fewer parameters than SAM variants. The efficiency gains translate to faster inference times (7b), with FLIP-Large requiring only 44.13ms compared to SAM2-L's 88.64ms, representing an approximate $2\times$ speedup alongside improved accuracy (measured on a Nvidia RTX4090 graphics card).

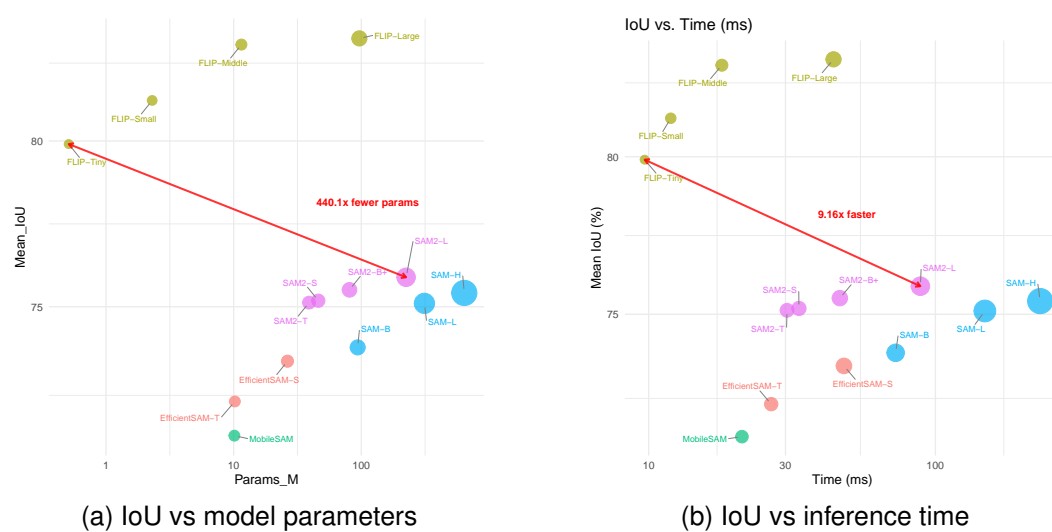

(a) IoU vs model parameters      (b) IoU vs inference time

Figure 7: Comparison of mean IoU across six diverse datasets (Hypersim, KITTI-360, OpenImages, COCO, LVIS, ObjaScale) plotted against (a) model size (in millions of parameters) and (b) inference time (in milliseconds). FLIP variants consistently outperform SAM variants while using several orders of magnitude fewer parameters and being significantly faster.

## 5 CONCLUSION

We have introduced FLIP, a novel vision model that fundamentally rethinks object segmentation through fovea-inspired input patching. By directly sampling multi-resolution patches centered on objects of interest, FLIP achieves state-of-the-art segmentation performance while using orders of magnitude fewer parameters than existing approaches.

Our key innovation lies in the scale-invariant patch sampling mechanism that adapts processing resolution to object characteristics, allocating computational resources where they matter most. This design enables FLIP-Large to achieve 83.26% mean IoU with only 96.6M parameters—outperforming SAM2-L's 75.87% IoU (224.45M parameters) while being about $2\times$ faster. Even our smallest model, FLIP-Tiny (0.51M parameters), surpasses all SAM variants with 79.90% mean IoU.

The introduction of ObjaScale reveals a critical limitation in current segmentation models: their inability to effectively handle very small objects in high-resolution scenes. FLIP addresses this gap through its fovea-like sampling, maintaining 88.6–90.7% IoU on ObjaScale where SAM variants achieve only 67.7–73.8%. This capability, combined with consistent performance improvements across Hypersim, KITTI-360, OpenImages, COCO, and LVIS, demonstrates FLIP's potential for applications requiring precise segmentation of objects across extreme scale variations.

Looking forward, FLIP's architectural efficiency and strong generalization open new possibilities for real-time object-centric vision systems. Our current formulation, however, assumes access to a 2D Gaussian prompt, which is not yet a drop-in replacement for the bounding-box- or point-based prompts used in most deployed detection and segmentation pipelines, so deriving such Gaussians from detector boxes, tracking filters, or slot-based scene models and studying robustness to imperfect, non-Gaussian prompts remains important future work. We nevertheless expect FLIP's foveated encoder and Gaussian object prompts to be particularly useful in tracking scenarios where an additional temporal module may predict future 2D Gaussian locations of objects across frames. Our results suggest that biologically inspired selective attention mechanisms can deliver both superior accuracy and computational efficiency, paving the way for more capable and sustainable computer vision applications in domains ranging from autonomous navigation to medical imaging.

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

# A APPENDIX

## A.1 ENCODER AND PIXEL-LEVEL MASK PREDICTION

### A.1.1 ENCODER

For encoding the multi-resolution patches, we employ a Vision Transformer architecture with modified attention blocks inspired by LaPE (Yu et al., 2023). Our design differs from standard transformers by incorporating position-aware attention without position embedding normalization, maintaining separate layer normalization for queries, keys, and values, which relates to recent analyses of how normalization placement affects semantic subspaces in transformers (Menary et al., 2024).

8b illustrates the architecture of our attention blocks. Each block receives the set of tokens $X_l$ from the previous layer as input. The first layer receives $X_1 = T$, which are the tokens produced by patch-size-specific embedding MLPs $E_{r_0}..E_{r_K}$ that process the foveal patches. Similar to LaPE, we inject relative positional embeddings PE into every layer.

The relative positional embeddings $(\hat{x}_i, \hat{y}_i)$ are computed with respect to the 2D Gaussian input query. From $\Sigma$, we extract rotation $(\theta_a, \theta_b)$ and scale $(\sigma_x, \sigma_y)$ via a standard eigenvalue decomposition of the covariance matrix. For each token at position $(x_i, y_i)$, we shift coordinates by $(\mu_x, \mu_y)$, rotate by $(\theta_a, \theta_b)$, and scale by $(\sigma_x, \sigma_y)$. We then use separate MLPs to compute positional embeddings for queries, keys and values:

$$\text{PE}_i^{q,k,v} = \text{MLP}_i^{q,k,v}(\hat{x}_i, \hat{y}_i),$$
$$\mathbf{Q}_i' = \mathbf{Q}_i + \text{PE}_i^q, \quad \mathbf{K}_i' = \mathbf{K}_i + \text{PE}_i^k, \quad \mathbf{V}_i' = \mathbf{V}_i + \text{PE}_i^v,$$

where MLP is a two-layer MLP with SiLU activations that maps 2D coordinates to positional embeddings.

Unlike standard transformers, we apply individual layer normalization to queries, keys, and values before projection, which empirically improved performance over a single layer normalization (see Table 3):

$$Q = \text{LN}_Q(X_l)W_Q, \quad K = \text{LN}_K(X_l)W_K, \quad V = \text{LN}_V(X_l)W_V,$$

where $\text{LN}_Q$, $\text{LN}_K$, and $\text{LN}_V$ are separate LayerNorm modules.

The attention computation proceeds as:

$$Q' = Q + \text{PE}_Q, \quad K' = K + \text{PE}_K, \quad V' = V + \text{PE}_V,$$

$$\text{Attn}(Q', K', V) = \text{softmax}\left(\frac{Q'K'^T}{\sqrt{d_k}}\right)V',$$

$$Y = X_l + \alpha \cdot \text{Attn}(Q', K', V')W_{out},$$

where $\alpha$ is a learnable scaling parameter initialized to 0.1 for stable training, and $d_k$ is the key dimension.

Each attention block is followed by a residual MLP:

$$X_{l+1} = Y + \beta \cdot \text{MLP}(\text{LN}(Y)),$$

where $\beta$ is another learnable scaling parameter. The MLP uses a bottleneck design with SiLU activations and expansion factor of 4. Depending on the model configuration, we stack 3, 5, 8, or 24 of these blocks to form the complete encoder.

### A.1.2 PIXEL-LEVEL MASK PREDICTION

The Pixel-Predictor generates precise segmentation masks by computing attention between encoded patch features and query pixels. Given the encoder output tokens $X_e$, we predict mask values for arbitrary pixel coordinates through a sparse attention mechanism.

The predictor consists of preprocessing and postprocessing residual MLP layers surrounding a specialized attention module. We first preprocess the encoder features:

$$\hat{X}_e = \text{MLP}_{\text{pre}}(X_e)$$

For each target pixel at coordinates $(x_t, y_t)$, we compute relative positional embeddings with respect to the input 2D Gaussian by shifting coordinates by $(\mu_x, \mu_y)$, rotating by $(\theta_a, \theta_b)$, and scaling by $(\sigma_x, \sigma_y)$ to get $(\hat{x}_t, \hat{y}_t)$:

$$\text{PE}_t = \text{MLP}_t(\hat{x}_t, \hat{y}_t),$$
$$\text{PE}_i = \text{MLP}_i(\hat{x}_i, \hat{y}_i),$$

where MLP is a two-layer MLP with SiLU activations that maps 2D coordinates to positional embeddings, and $\text{PE}_i$ represents the positional embeddings for each encoded and preprocessed input patch $(\hat{X}_e)$.

In contrast to the encoder, the attention mechanism in the Pixel-Predictor is not residual. We compute embedding-informed keys and values as follows:

$$K_{\text{enhanced}} = \text{MLP}_K(\text{concat}[\text{LN}_K(\hat{X}_e), \text{PE}_i]),$$
$$V_{\text{enhanced}} = \text{LN}_V(\hat{X}_e)W_V + \text{MLP}_V(\text{PE}_i)$$

For each query pixel, we compute:

$$Q_{\text{pixel}} = \text{PE}_t,$$
$$\text{feat}_{\text{pixel}} = \text{Attention}(Q_{\text{pixel}}, K_{\text{enhanced}}, V_{\text{enhanced}}),$$
$$M_{x,y} = \text{Linear}(\text{feat}_{\text{pixel}} + \text{MLP}_{\text{post}}(\text{feat}_{\text{pixel}}))$$

where $M_{x,y}$ is the predicted mask logit at position $(x, y)$.

### A.2 SMALL OBJECT SEGMENTATION

Table 2 reveals FLIP's capability for small object segmentation (objects $<1\%$ image area). FLIP-Large achieves 81.19% mean IoU on small objects compared to SAM2-L's 73.50%, demonstrating the effectiveness of our fovea-like patching mechanism. This advantage is most pronounced on ObjaScale, where FLIP-Tiny achieves 88.28% IoU on small objects compared to SAM-H's 72.26% or SAM2-L's 67.70%.

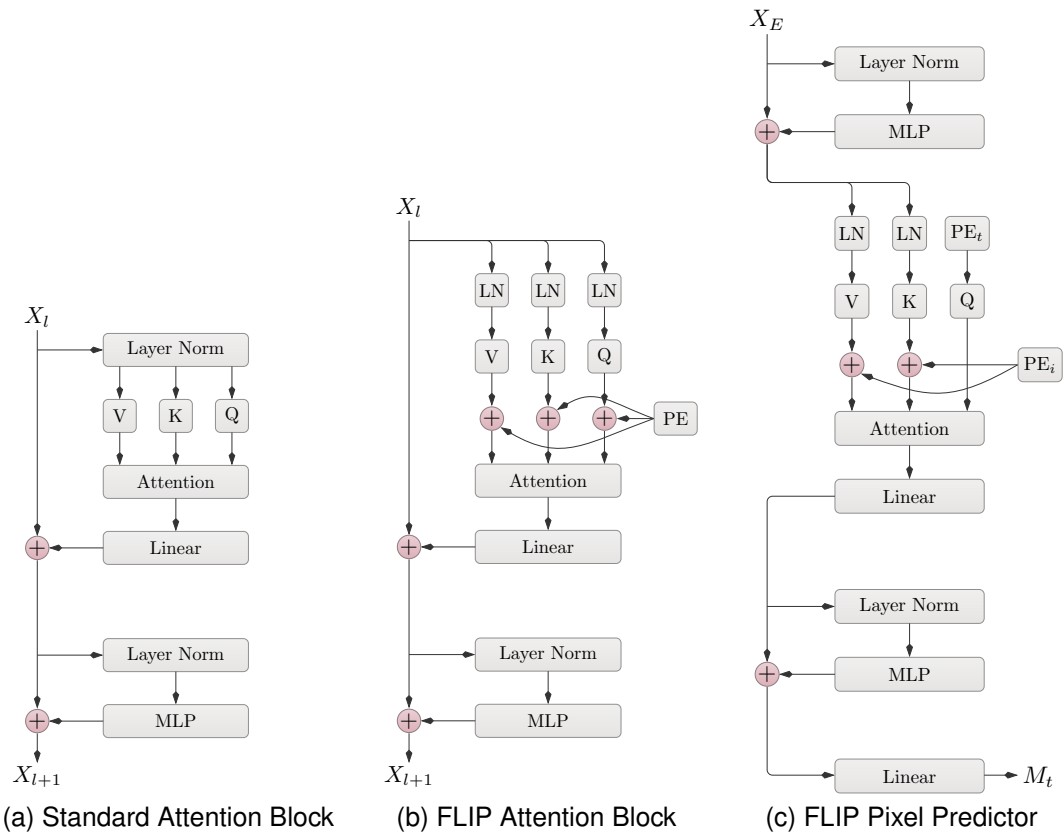

Figure 8: Architectural components of FLIP. (a) Standard pre-layer norm attention block for comparison. (b) Architecture of the FLIP attention block, where positional embeddings influence queries, keys and values independent of feature normalization. (c) Architecture of the FLIP Pixel-Predictor. Encoder features $X_E$ are processed through a residual MLP. The attention mechanism computes keys (K) and values (V) enhanced with patch positional embeddings ($PE_i$), while queries (Q) use positional embeddings derived from the target pixels coordinates ($PE_t$). The attended features pass through a residual MLP and a final linear layer to produce mask predictions $M_t$ for each query pixel.

The heat-map analysis (Figure 6) offers clear evidence of FLIP's scale-invariant behavior: whereas SAM variants—and FastSAM-x in particular—suffer sharp performance drops when the mask-to-image ratio falls below 0.01%, FLIP's IoU remains stable until the object's absolute footprint shrinks to roughly $10 \times 10$ pixels. Because FLIP's accuracy is governed by absolute pixel coverage rather than relative image area, it robustly segments small objects even in ultra-high-resolution scenes—a critical advantage for real-world applications.

## A.3 ARCHITECTURAL ROBUSTNESS AND ABLATION ANALYSIS

Our ablation study (Table 3) reveals that FLIP is remarkably robust to architectural variations while certain design choices provide measurable improvements. Unless stated otherwise, all ablations are evaluated without any prompt-specific fine-tuning. The baseline FLIP-Small (79.29% mean IoU) serves as our reference point for evaluating architectural components.

**Positional Embedding Strategies.** The comparison between baseline and "InitialPosEmb" (79.23% vs. 79.29%) shows that our layer-wise positional embedding injection provides marginal but consistent benefits over initial embedding only. However, the "LaPE-SNorm" variant, which applies shared normalization across positional embeddings, achieves the best performance at 79.50% mean IoU, while underperforming in the hierarchical inference setting.

Table 2: Segmentation Accuracy on small-scale objects ($< 1\%$ image area)

| Model | Size (M) | Time (ms) | Hypersim | KITTI-360 | OpenImage | COCO | LVIS | ObjaScale (ours) | Average |
|---|---|---|---|---|---|---|---|---|---|
| | | | Mean ± Std IoU (%) | Mean ± Std IoU (%) | Mean ± Std IoU (%) | Mean ± Std IoU (%) | Mean ± Std IoU (%) | Mean ± Std IoU (%) | Mean ± Std IoU (%) |
| FastSAM-s | 11.8 | 10.02 | 31.91 ± 32.85 | 32.55 ± 31.32 | 50.59 ± 32.66 | 38.50 ± 32.74 | 31.12 ± 33.01 | 45.91 ± 35.16 | 38.43 ± 32.96 |
| FastSAM-x | 72.2 | 24.37 | 24.77 ± 30.20 | 33.12 ± 33.34 | 56.58 ± 31.82 | 44.09 ± 34.05 | 34.96 ± 35.00 | 44.13 ± 35.62 | 39.61 ± 33.34 |
| MobileSAM | 10.13 | 21.83 | 65.67 ± 20.77 | 62.36 ± 18.41 | 76.76 ± 19.01 | 70.19 ± 17.36 | 68.46 ± 21.25 | 65.91 ± 24.86 | 68.22 ± 20.28 |
| EfficientSAM-T | 10.22 | 26.75 | 68.61 ± 20.18 | 63.11 ± 17.56 | 79.67 ± 16.78 | 72.91 ± 16.41 | 72.79 ± 19.80 | 67.11 ± 26.17 | 70.70 ± 19.48 |
| EfficientSAM-S | 26.41 | 47.98 | 66.74 ± 21.33 | 63.64 ± 18.01 | 80.82 ± 17.48 | 73.26 ± 16.78 | 71.64 ± 21.16 | 65.86 ± 26.63 | 70.33 ± 20.23 |
| SAM-B | 93.7 | 73.33 | 69.97 ± 19.58 | 63.05 ± 18.28 | 81.08 ± 16.87 | 74.48 ± 15.60 | 74.53 ± 19.11 | 69.79 ± 25.41 | 72.15 ± 19.14 |
| SAM-L | 312.3 | 149.45 | 69.76 ± 20.29 | 62.40 ± 18.44 | 82.76 ± 15.33 | 75.39 ± 15.59 | 74.75 ± 19.71 | 71.12 ± 25.32 | 72.70 ± 19.11 |
| SAM-H | 641.1 | 232.67 | 70.12 ± 20.66 | 62.02 ± 18.50 | 82.66 ± 15.81 | 75.62 ± 15.63 | 75.18 ± 19.83 | 72.26 ± 24.71 | 72.98 ± 19.19 |
| SAM2-T | 38.96 | 31.37 | 70.08 ± 18.88 | 67.29 ± 17.34 | 82.91 ± 14.38 | 75.65 ± 14.78 | 73.68 ± 19.44 | 68.09 ± 26.27 | 72.95 ± 18.51 |
| SAM2-S | 46.06 | 34.45 | 70.23 ± 18.89 | 67.63 ± 17.58 | 83.47 ± 14.23 | 75.91 ± 14.81 | 73.89 ± 19.51 | 67.87 ± 26.39 | 73.17 ± 18.57 |
| SAM2-B+ | 80.85 | 47.43 | 70.13 ± 19.08 | 65.60 ± 17.15 | 83.69 ± 13.73 | 76.07 ± 14.83 | 73.81 ± 19.66 | 68.72 ± 26.55 | 73.00 ± 18.50 |
| SAM2-L | 224.45 | 89.61 | 70.95 ± 18.88 | 67.98 ± 17.56 | 84.48 ± 12.70 | 76.43 ± 14.49 | 73.48 ± 19.86 | 67.70 ± 26.99 | 73.50 ± 18.41 |
| FLIP-Tiny | **0.51** | 8.50 | 71.48 ± 21.90 | **70.47** ± 16.86 | 84.39 ± 14.21 | 78.22 ± 14.93 | 75.25 ± 20.70 | 88.28 ± 15.33 | 78.01 ± 17.32 |
| FLIP-Small | 2.3 | 10.35 | 72.41 ± 21.77 | 69.99 ± 16.76 | 86.56 ± 11.92 | 79.19 ± 14.93 | 76.32 ± 21.20 | 89.38 ± 15.21 | 78.97 ± 16.96 |
| FLIP-Middle | 11.5 | 13.89 | 76.12 ± 20.57 | 68.82 ± 16.51 | 88.44 ± 10.21 | 81.15 ± 13.31 | 79.98 ± 18.69 | **90.46** ± 14.10 | 80.83 ± 15.56 |
| FLIP-Large | 96.6 | 30.52 | **77.21** ± 20.15 | 67.58 ± 17.21 | **89.14** ± 8.57 | 80.52 ± 14.30 | **81.24** ± 17.89 | 90.45 ± 14.00 | **81.19** ± 15.18 |
| FLIP$_{bb}$-Tiny | **0.51** | **8.20** | 66.86 ± 21.35 | 65.36 ± 18.14 | 80.29 ± 15.55 | 74.82 ± 15.86 | 73.81 ± 20.04 | 86.52 ± 16.68 | 74.61 ± 17.94 |
| FLIP$_{bb}$-Small | 2.3 | 9.85 | 68.47 ± 21.02 | 65.72 ± 17.93 | 82.76 ± 13.93 | 76.58 ± 15.08 | 75.79 ± 19.38 | 88.10 ± 16.30 | 76.20 ± 17.27 |
| FLIP$_{bb}$-Middle | 11.5 | 12.37 | 69.50 ± 20.76 | 65.56 ± 18.11 | 84.37 ± 12.50 | 77.82 ± 14.64 | 77.41 ± 18.68 | 88.60 ± 15.96 | 77.21 ± 16.78 |
| FLIP$_{bb}$-Large | 96.6 | 25.57 | 70.04 ± 20.61 | 65.29 ± 18.60 | 85.48 ± 11.45 | 78.22 ± 14.60 | 78.23 ± 18.19 | 88.60 ± 15.95 | 77.64 ± 16.57 |

**Patch Resolution Configurations.** Testing different patch size ranges reveals interesting trade-offs. Restricting maximum patch size to $4\times4$ ("MaxPatch4x4") reduces parameters to 1.7M with only modest performance degradation (78.81% IoU), demonstrating FLIP's efficiency even with limited resolution diversity. Conversely, extending to $64\times64$ patches ("MaxPatch64x64") slightly improves performance (79.44%) at the cost of increased parameters (3.0M), suggesting diminishing returns from very large patches.

Removing the smallest $1\times1$ patches ("MinPatch4x4") significantly impacts performance on LVIS (72.08% vs. 77.92%), highlighting the importance of fine-grained detail capture for complex segmentation tasks, while maintaining strong performance on other datasets.

**Mask Target Pixel Sampling Strategy Impact.** The "NoDynamicSampling" ablation (78.63% vs. 79.29%) demonstrates that our adaptive target pixel sampling strategy provides consistent benefits across datasets. Without dynamic sampling, each distance group uses the same number of pixels rather than adaptively redistributing based on prediction difficulty. The performance gap is particularly notable on Hypersim (71.03% vs. 73.07%), where focusing training on challenging boundary regions better handles the diverse object shapes in simulated indoor scenes.

## A.4 CROSS-DATASET GENERALIZATION

FLIP demonstrates robust generalization across diverse evaluation benchmarks without dataset-specific fine-tuning. The performance gains are consistent across synthetic (Hypersim), automotive (KITTI-360), and large-scale natural image datasets (OpenImages, COCO, LVIS). Notably, FLIP-Tiny achieves 72.62% IoU on KITTI-360 compared to SAM-H's 62.47%, despite using $1{,}257\times$ fewer parameters. This 10.15 percentage point improvement highlights how FLIP's fovea-like attention mechanism effectively handles automotive scenarios characterized by numerous small, distant objects.

Table 3: FLIP-Small ablation study

| Ablation | Params (M) | Time (ms) | Hypersim | KITTI-360 | OpenImages | COCO | LVIS | ObjaScale (ours) | Average |
|---|---|---|---|---|---|---|---|---|---|
| | | | Mean ± Std | Mean ± Std | Mean ± Std | Mean ± Std | Mean ± Std | Mean ± Std | Mean ± Std |
| Baseline | 2.3 | 12.19 | 73.07 ± 19.80 | 69.91 ± 15.69 | 86.65 ± 11.27 | 79.23 ± 13.32 | 77.92 ± 18.34 | 88.96 ± 15.14 | 79.29 ± 15.59 |
| Baseline$_h$ | 2.3 | 10.94 | 71.93 ± 18.99 | 68.34 ± 15.57 | 85.73 ± 11.29 | 78.15 ± 13.04 | 76.71 ± 17.41 | 86.97 ± 14.68 | 77.97 ± 15.16 |
| InitialPosEmb | 2.2 | 10.37 | 73.41 ± 19.73 | 68.96 ± 15.48 | 86.29 ± 11.54 | 79.22 ± 13.14 | **78.42** ± 17.98 | 89.05 ± 14.98 | 79.23 ± 15.47 |
| InitialPosEmb$_h$ | 2.2 | 8.98 | 72.03 ± 18.92 | 67.36 ± 15.27 | 85.45 ± 11.51 | 77.94 ± 12.98 | 76.92 ± 17.15 | 86.84 ± 14.63 | 77.76 ± 15.08 |
| LaPE-SNorm | 2.3 | 10.88 | **73.86** ± 19.37 | 69.92 ± 15.20 | 86.41 ± 11.40 | 79.29 ± 13.12 | 78.42 ± 18.00 | **89.12** ± 14.77 | **79.50** ± 15.31 |
| LaPE-SNorm$_h$ | 2.3 | **8.79** | 71.03 ± 17.74 | 65.87 ± 13.78 | 80.20 ± 11.98 | 74.42 ± 12.71 | 74.34 ± 16.30 | 83.01 ± 14.61 | 74.81 ± 14.52 |
| MaxPatch4x4 | **1.7** | 11.46 | 72.69 ± 19.71 | 69.43 ± 15.57 | 86.01 ± 11.55 | 78.73 ± 13.28 | 77.55 ± 18.30 | 88.48 ± 14.96 | 78.81 ± 15.56 |
| MaxPatch4x4$_h$ | **1.7** | 10.09 | 71.45 ± 18.97 | 67.87 ± 15.46 | 85.11 ± 11.47 | 77.61 ± 13.07 | 76.31 ± 17.43 | 86.55 ± 14.57 | 77.48 ± 15.16 |
| MaxPatch64x64 | 3.0 | 12.54 | 73.45 ± 19.52 | 69.78 ± 15.93 | 86.67 ± 11.37 | **79.33** ± 13.15 | 78.39 ± 18.02 | 89.04 ± 14.89 | 79.44 ± 15.48 |
| MaxPatch64x64$_h$ | 3.0 | 11.10 | 72.19 ± 18.80 | 68.21 ± 15.80 | 85.73 ± 11.32 | 78.16 ± 12.91 | 77.04 ± 17.08 | 87.05 ± 14.41 | 78.06 ± 15.05 |
| MinPatch4x4 | 2.2 | 11.45 | 69.39 ± 23.77 | **70.30** ± 15.60 | 86.42 ± 11.60 | 77.27 ± 16.62 | 72.08 ± 25.85 | 88.14 ± 17.28 | 77.27 ± 18.45 |
| MinPatch4x4$_h$ | 2.2 | 10.07 | 68.53 ± 22.58 | 68.82 ± 15.46 | 85.53 ± 11.60 | 76.31 ± 16.23 | 71.07 ± 24.74 | 86.05 ± 16.78 | 76.05 ± 17.90 |
| NoDynamicSampling | 2.3 | 12.17 | 71.03 ± 20.14 | 70.03 ± 15.46 | **87.47** ± 11.07 | 79.04 ± 13.65 | 76.40 ± 19.00 | 87.82 ± 15.95 | 78.63 ± 15.88 |
| NoDynamicSampling$_h$ | 2.3 | 10.69 | 70.57 ± 19.46 | 68.54 ± 15.26 | 86.63 ± 11.07 | 78.29 ± 13.38 | 75.89 ± 18.11 | 86.66 ± 15.20 | 77.76 ± 15.41 |

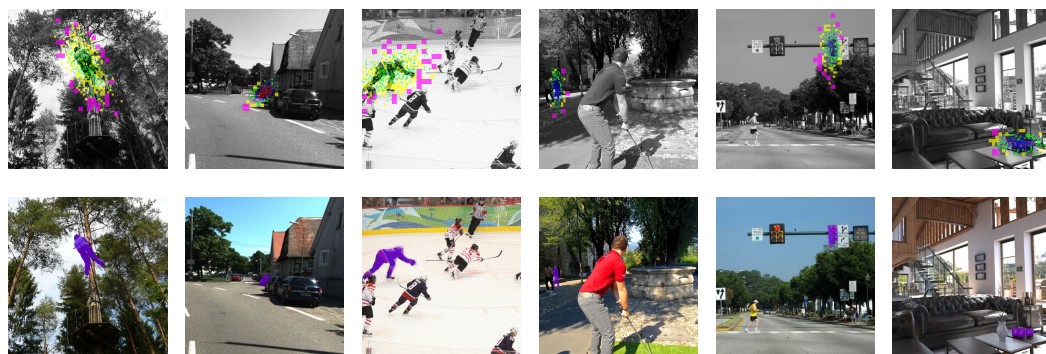

Figure 9: Examples of multi-resolution patch inputs (top row) and corresponding mask predictions (bottom row) from FLIP. Input patches are color-coded by size: purple ($16 \times 16$), yellow ($8 \times 8$), green ($4 \times 4$), blue ($2 \times 2$), and red ($1 \times 1$). Higher-resolution patches focus on object centers for detail, while lower-resolution patches cover peripheral regions for efficiency. Mask predictions show accurate segmentation with optimized resource allocation.

Table 4: Non-finetuned 2d Gaussian prompted FLIP models (standard and hierarchical).

| Model | Size (M) | Time (ms) | Hypersim | KITTI-360 | OpenImage | COCO | LVIS | ObjaScale (ours) | Average |
|---|---|---|---|---|---|---|---|---|---|
| | | | Mean ± Std IoU (%) | Mean ± Std IoU (%) | Mean ± Std IoU (%) | Mean ± Std IoU (%) | Mean ± Std IoU (%) | Mean ± Std IoU (%) | Mean ± Std IoU (%) |
| FLIP-Tiny | **0.51** | 9.82 | 73.17 ± 19.36 | **70.04** ± 15.27 | 83.83 ± 12.72 | 77.76 ± 13.55 | 76.88 ± 18.33 | 87.76 ± 14.93 | 78.24 ± 15.69 |
| FLIP-Small | 2.3 | 12.19 | 73.07 ± 19.80 | 69.91 ± 15.69 | 86.65 ± 11.27 | 79.23 ± 13.32 | 77.92 ± 18.34 | 88.96 ± 15.14 | 79.29 ± 15.59 |
| FLIP-Middle | 11.5 | 17.54 | **73.68** ± 20.13 | 69.86 ± 15.76 | 87.89 ± 10.08 | 80.16 ± 13.15 | 78.88 ± 18.31 | **89.08** ± 15.04 | 79.93 ± 15.41 |
| FLIP-Large | 96.6 | 38.65 | 73.66 ± 20.51 | 69.75 ± 16.03 | **89.33** ±  8.76 | **80.79** ± 13.06 | **79.42** ± 18.13 | 89.02 ± 15.21 | **80.33** ± 15.28 |
| FLIP$_h$-Tiny | **0.51** | **9.05** | 71.36 ± 18.65 | 68.53 ± 15.08 | 83.03 ± 12.65 | 76.57 ± 13.25 | 75.41 ± 17.37 | 85.72 ± 14.52 | 76.77 ± 15.25 |
| FLIP$_h$-Small | 2.3 | 10.94 | 71.93 ± 18.99 | 68.34 ± 15.57 | 85.73 ± 11.29 | 78.15 ± 13.04 | 76.71 ± 17.41 | 86.97 ± 14.68 | 77.97 ± 15.16 |
| FLIP$_h$-Middle | 11.5 | 14.34 | 72.65 ± 19.21 | 68.28 ± 15.68 | 86.98 ± 10.01 | 78.98 ± 12.89 | 77.58 ± 17.25 | 86.95 ± 14.68 | 78.57 ± 14.95 |
| FLIP$_h$-Large | 96.6 | 29.36 | 73.19 ± 19.56 | 68.05 ± 16.18 | 88.28 ±  8.96 | 79.67 ± 12.83 | 78.40 ± 17.10 | 87.36 ± 14.61 | 79.16 ± 14.87 |

The architectural ablations confirm that FLIP's performance is remarkably robust to design variations. While removing the smallest patches (MinPatch4x4) impacts performance on complex datasets like LVIS, and the adaptive pixel sampling during training (NoDynamicSampling ablation) provides consistent improvements, the overall architecture maintains strong performance across different configurations. This robustness suggests that FLIP's core fovea-like attention mechanism, rather than any single architectural choice, drives its superior accuracy, particularly for challenging small objects.

## A.5 HIERARCHICAL REFINEMENT

**Procedure**    For further efficiency, we introduce a hierarchical inference strategy that progressively refines a 5-sigma bounding-box mask prediction. Starting from a coarse resolution, we predict initial mask values on a coarse grid and identify uncertain regions where predictions fall within $[\tau, 1 - \tau]$ (default $\tau = 0.01$). The mask is then upsampled by a factor $\alpha$ (default $\alpha = 4$) using bilinear interpolation, but only the uncertain pixels are re-queried at the higher resolution. This process repeats iteratively until the target resolution is reached, focusing refinement exclusively on ambiguous boundary regions. The number of refinement steps $n$ is automatically determined as $n = \lfloor \log_\alpha(k_{\text{target}}/k_{\text{init}}) \rfloor$, where $k_{\text{init}}$ and $k_{\text{target}}$ are the initial and target resolutions.

**Empirical Trade-offs.**    The hierarchical inference variants (FLIP$_h$) demonstrate the effectiveness of our progressive refinement strategy, achieving comparable performance with reduced computational cost. Comparing standard inference with hierarchical variants reveals a general pattern: hierarchical inference achieves 98–99% of the original performance while providing $\approx$ 8–24% speed improvements. For instance, FLIP$_h$-Large maintains 79.16% mean IoU while reducing inference time to 29.36ms. This trade-off is favorable for real-time applications where slight accuracy reductions are acceptable for substantial computational savings.

**Non-finetuned Gaussian Variants.**    Table 4 summarizes the Gaussian-query FLIP and FLIP$_h$ variants. These results reuse the checkpoints from the main training before prompt-specific finetuning, isolating the effect of the progressive inference scheme.

### A.6 LLM USAGE DISCLOSURE.

This paper is the authors' original work. During writing, we used AI language models to draft initial phrasing and rephrase sections for clarity and coherence. All AI-assisted text was reviewed, revised, and validated by the authors; all ideas, methods, experiments, analyses, and claims are solely by the authors.

