# OpenReview forum: "Looking Locally: Object-Centric Vision Transformers as Foundation Models for Efficient Segmentation"
_ICLR.cc/2026/Conference — Submitted to ICLR 2026_

### Official Review · Reviewer_FDjP · 2025-10-17

**Soundness:** 3
**Presentation:** 3
**Contribution:** 3
**Rating:** 8
**Confidence:** 4

**Summary:**

The paper introduces FLIP (Fovea-Like Input Patching), a parameter-efficient model for object segmentation.
Inspired by biological foveation, FLIP adaptively samples multi-resolution patches centered on objects of interest, allocating high-resolution processing to object centers while maintaining coarser peripheral context.
Key contributions include:
1. Off-grid and scale-invariant design for efficient small-object segmentation in high-resolution scenes.
2. Exceptional parameter efficiency: FLIP-Tiny (0.51M params) outperforms SAM-H (641M params) by 2.83% mIoU while running 23.6× faster.
3. State-of-the-art results on 6 benchmarks (e.g., 80.33% mIoU vs. SAM-H’s 75.41% on ObjaScale, a new dataset stress-testing scale invariance).

**Strengths:**

1. The fovea-like patching mechanism is an interesting idea. It introduces a biologically plausible, attention-based sampling strategy.
2. The ObjaScale dataset is a valuable contribution, filling a gap in benchmarking scale-invariant segmentation, especially for very small objects.
3. The parameter efficiency is remarkable. FLIP-Tiny (0.51M params) beats SAM-H (641M params) in mean IoU.
4. The speedup (6× faster than SAM-H) is significant and practically relevant.

**Weaknesses:**

1. No mention of code release or reproducibility checklist. For ICLR, open-sourcing code and models is strongly expected, especially when claiming efficiency gains.
2. While SAM and its variants are strong baselines, the paper does not compare against other efficient segmentation models. All models are from 2023, no models in 2024 and 2025.
3. The ObjaScale dataset is synthetic and Blender-rendered. While useful, it may not reflect real-world complexity (e.g., motion blur, occlusion, and lighting variation).

**Questions:**

No

---

> ### Author Response · Authors · 2025-11-17
> **Answer to Reviewer FDjP**
>
> We thank Reviewer FDjP for the constructive feedback and positive evaluation of our paper. We want to mention that the source-code is provided in the supplementary material and will be made open source. Also we now do compare against SAMv2 and FLIP-Tiny still outperforms SAM2-L on average.

---

### Official Review · Reviewer_XpqH · 2025-10-21

**Soundness:** 3
**Presentation:** 3
**Contribution:** 2
**Rating:** 4
**Confidence:** 4

**Summary:**

This paper proposes FLIP for object-centric segmentation. FLIP samples multi resolution patches around a Gaussian prompt, encodes them with a ViT using position aware attention, and predicts masks via sparse pixel queries. Experiments on various datasets show reasonable accuracy and efficiency with fewer parameters and lower runtime, including stable performance on small objects.

**Strengths:**

1. The paper is well written, and the motivation is clear and convincing.

2. FLIP provides an effective way to reduce the patch count while preserving segmentation accuracy, and it includes an efficient low level implementation of fovea patching.

3. The experimental results demonstrate reasonable and consistent performance gains across datasets.

**Weaknesses:**

1. The fovea inspired input patching appears incremental. STT [1] presents a similar idea to speed up SAM. The task setting is very close. Both use a prompt to focus on the object and build multi level foveated patches. FLIP relies on less structured and partly random sampling. STT uses a more structured tokenization. The paper does not cite STT. STT is CVPR 2025 and is within scope for ICLR 2026. This omission weakens the novelty claim.

2. The comparison may be unfair. FLIP uses a Gaussian derived from the GT mask. SAM and its variants (EfficientSAM, MobileSAM, FastSAM) use a GT bounding box. These prompts provide different information. This mismatch can bias results. To ensure fairness, the evaluation should use the same input across SAM-family baselines and FLIP. A reasonable protocol is to use the GT box for all SAM-family methods and convert the box to a Gaussian for FLIP within preprocessing, then compare. This setup also aligns with practical use with detector generated boxes.

3. The paper does not provide enough detail on how the fovea patches are constructed, which harms understanding of the core contribution of this work.

[1] Segment This Thing: Foveated Tokenization for Efficient Point Prompted Segmentation, CVPR 2025.

**Questions:**

Please refer to the Weaknesses section. This paper has some technical contribution, but STT has already presented a similar idea. Considering the fairness issue in the comparison, only the current score can be given at this time.

---

> ### Author Response · Authors · 2025-11-17
> **Answer to Reviewer XpqH**
>
> We thank Reviewer XpqH for the constructive feedback. Below we provide a TLDR and also our point-by-point responses.
>
> ---
> ## TLDR:
> STT is concurrent work and differs fundamentally (point-prompted grid downsampling vs. our off-grid Gaussian sampling); we now cite and compare against it, retain model-optimal prompts as the fairest protocol, and Section 3.1 plus source-code detail the patch-sampling.
>
> ---
> ## Novelty issues:
> We where not aware of the STT paper, but have cited and evaluated it in our latest revision. We want to stress the point that STT and FLIP where developed concurrently, the arXiv timestep of STT is 10 Jun 2025 which is only around 2 Month before the ICLR submission deadline.
>
> Apart from this, it needs to be pointed out that STT does outperform Sam/Efficient Sam ONLY on ONE of the nine reported datasets (two ties, six times SAM-H/SAM-L or EfficientSAM S win - partially with relatively large margins, cf. Table 5 in STT paper)!
> Our FLIP system outperforms SAM/Efficient SAM on ALL of the six reported datasets.
>
>
> Further, FLIP is by no means incremental when compared to STT.
> While FLIP and STT are both inspired by foveation, they approach the single object segmentation vastly different: STT starts with a point prompt and divides a cropped input grid centered on this point into increasingly bigger patches. The starting minimum patch size of STT is 16x16 and all bigger patches are down-sampled to this 16x16 patch size. In contrast our FLIP model extracts patches that don't lie on a grid but are sampled from a real valued 2d gaussian distribution which is our input prompt. The sampled patches in FLIP are then extracted from the image exactly at the sampled patch position using bilinear interpolation from the image pixel grid to the patch coordinates. Additionally, our maximum patch size is 16x16 and our smallest patch size is 1x1 in our ablations, we specifically show that using these smaller patch sizes drastically improves performance (see supplementary section of the paper).
>
>
> STT uses a much bigger transformer with 90.11M (STT-B), 307.67M (STT-L) and 635.34M (STT-H) parameters and their main speedup comes from using fewer tokens (around 23.8 times less than SAM), in contrast FLIPs main speedup comes from reduced parameter counts we only use 4x less tokens than SAM but our smallest model FLIP-T with 0.5M parameters (more than 1000 times less than SAM-H) already outperforms SAM-H on average (in 3 out of 6 datasets; SAM-B in 5 out of 6 datasets with still nearly 200 times less parameters).
>
>
> A fair comparison of our FLIP approach with STT is not directly possible as STT only uses point prompts. Point prompts provide much less information about the target object than bounding box prompts or 2d Gaussian prompts and are in fact ambiguous (which is clearly stated in both the STT and SAM papers). This is a big disadvantage for STT when compared to FLIP and can also be seen in the SAM evaluations from the STT paper. The reported scores are in general MUCH LOWER (for both, SAM and its variants as well as the STT approach) than in our bounding box prompt setting. Nevertheless, we ran STT through our evaluation pipeline and adopted the same point prompt selection as used in the STT paper (selecting the point inside the mask that is the farthest away from any mask boundary). As expected STT performed much worse in comparison to the other models that use unambiguous bounding box or 2d Gaussian prompts. The only exception is ObjaScale where STT performs comparable to SAM V1 and actually outperforms SAM V2 but is still nowhere near the performance of FLIP.
>
> ---
> ## Prompt mismatch:
>  - We acknowledge the difference in the prompting style of SAM variants and FLIP. Each give the model different object cues: 2d Gaussians give more direct shape cues, while bounding boxes give more direct object boundary cues.
>  - Prompting FLIP with 2d Gaussians derived from bounding boxes will result in out-of distribution prompts, similarly to how prompting SAM variants with points sampled from 2d Gaussians will result in out of distribution behavior.
>  - We adopted the prompting strategy that gives the best performance for each model in this case bounding boxes for SAM-variants and 2d Gaussians for FLIP. We think this is a fair comparison and hope that the reviewer will acknowledge that two different methods can not be compared 100\% fair.
>
> ---
> ## Fovea patch sampling:
> The sampling of our foveated patches is described in detail in Section 3.1 We also provide the source-code in our Supplementary material, so anyone can fully understand all technical details that can not be outlined in a paper with a page limit.

---

> > ### Comment · Reviewer_XpqH · 2025-11-17
> >
> > Thank you for the detailed rebuttal and clarifications.
> >
> > According to the ICLR reviewer guide FAQ, papers are considered contemporaneous if they are published at a peer‑reviewed venue within two months before the full paper deadline (i.e., on or after July 24, 2025). STT’s arXiv timestamp is June 10, meaning it was publicly available well before the ICLR deadline and does not fall into this “last two months” window. In this context, once STT is brought into the discussion, it naturally serves as an important reference when evaluating the novelty and positioning of FLIP.
> >
> > The difference in prompting is also a central aspect of this evaluation. The rebuttal highlights that point prompts, as in STT, can be ambiguous, while Gaussian prompts provide richer shape cues. From an application perspective, however, such rich Gaussian prompts are usually harder to obtain reliably, whereas single-point prompts are comparatively easy to acquire (for example through interactive user input). STT already demonstrates this via its public demo, which shows a clear and realistic usage pattern. For FLIP, a concrete description of how such Gaussian prompts would be obtained in typical real-world pipelines has not been provided, which makes performance gains under Gaussian prompts harder to interpret as a practical advantage and also leaves open questions about fairness when comparing models under different levels of prompt information.
> >
> > The foveated patch construction is the main structural component that distinguishes FLIP from STT. Even if the main paper is space‑limited, the supplementary material could provide a more explicit and formally stated description of the sampling and construction process. A clearer and more detailed formulation there would help sharpen the core contribution and make the comparison with STT more transparent.
> >
> > Given these points, the rebuttal does not fully resolve the concerns about novelty, practical applicability, and fairness of comparison.

---

> > > ### Author Response · Authors · 2025-11-17
> > > **Answer to Reviewer XpqH**
> > >
> > > We respectfully disagree that FLIP’s novelty is exhausted by STT.
> > >
> > > First, the **core mechanism is different**: STT does grid-based tokenization around a point prompt on a SAM-scale encoder (all regions downsampled to 16×16), whereas FLIP uses **off-grid, continuous 2D Gaussian sampling with real-valued centers, 1×1–16×16 patches with overlap control, and a new lightweight ViT + pixel-query decoder with LaPE-style per-projection normalization and sparse boundary-focused supervision plus hierarchical 5σ inference**. STT only changes the tokenization of SAM; FLIP is a different architecture and operating regime.
> > >
> > > Second, **the empirical regime is different**: STT keeps 90–635M parameter backbones and STT does outperform Sam/Efficient Sam ONLY on ONE of the nine reported datasets (cf. Table 5 in STT paper)!, while FLIP-Tiny (0.51M params) already matches or surpasses SAM/SAM2, and FLIP-Large (96.6M) consistently outperforms SAM2-L across all six datasets. In our unified evaluation, STT (with its own optimal point-prompt protocol) is clearly weaker than FLIP, especially on ObjaScale.
> > >
> > > Finally, the **prompt type concerns evaluation fairness, not architectural novelty**. Using model-appropriate prompts (points for STT, Gaussians for FLIP, boxes for SAM) is orthogonal to whether FLIP’s off-grid foveation + pixel-query design is a substantive contribution, which we believe it is, and which the experiments support.

---

> > > > ### Comment · Reviewer_XpqH · 2025-11-17
> > > >
> > > > Thank you for the additional clarifications.
> > > >
> > > > Using a prompt centered region for foveated processing and efficiency is the core idea in both STT and FLIP. Another reviewer kR6o also noted this overlap. The rebuttal mainly highlights differences in how this idea is instantiated, but these look more like architectural and implementation choices. The encoder design is also not clearly framed as a conceptual innovation. At the same time, the paper does not provide a sufficiently clear and formal description of the foveated patch construction, which is central to FLIP. For an ICLR paper, this mechanism should be carefully formulated and illustrated so that readers can fully understand and verify the key technical contribution.
> > > >
> > > > On the application side and fairness of comparison, FLIP in effect defines a new task and benchmark. In that case, the paper needs to justify the meaning of this task, especially its connection to realistic scenarios. STT gives a more concrete picture here, as it explicitly targets streaming use cases such as robotics and AR or VR, and shows a gaze based prompting example, together with a public demo on the project page. I previously reviewed STT and found its description of application scenarios and prompting pipeline convincing. In contrast, FLIP currently constructs Gaussians directly from ground truth masks. This type of input does not exist in realistic pipelines and the paper does not propose a concrete application setting where such Gaussians would naturally arise. In the review I suggested deriving Gaussians from bounding boxes, which at least can be produced by standard detectors. If the method insists on Gaussians from ground truth masks, then in practice such precise masks would not be available and FLIP effectively assumes access to information that already solves a large part of the problem, while its robustness under inexact or noisy Gaussians is also not analyzed
> > > >
> > > > Given these points, the concerns raised in the original review are still not fully resolved

---

> > > > > ### Author Response · Authors · 2025-11-26
> > > > > **Answer to Reviewer XpqH**
> > > > >
> > > > > We thank the reviewer for their continued comments and time investment in the Rebuttal.
> > > > >
> > > > > We now also use bounding box prompts to evaluate FLIP. For this we fine-tuned our models to use 2d Gaussians derived from bounding box prompts. So essential the rotation component is set to a default value, while mu and sigma are directly computed from the bounding box. We acknowledge that this leads to a slightly lower performance, but want to highlight that FLIP-Tiny still outperforms SAM2-L on average with only 0.5M parameters compared to 224M for SAM2-L (see Table 1 in our latest revision).
> > > > >
> > > > > Additionally we followed Reviewer kR6o's suggestion and formalized our fovea like input patching more clearly in Algorithm 1 (see our latest revision). We also added two more figures to better visualize the patching process.
> > > > >
> > > > > Please also refer to our general comment to all reviewers summarizing the updated performance gains and comparisons.

---

### Official Review · Reviewer_jYnH · 2025-10-26

**Soundness:** 2
**Presentation:** 2
**Contribution:** 2
**Rating:** 4
**Confidence:** 3

**Summary:**

This paper introduces FLIP, an object-centric vision transformer that employs a fovea-like input patching mechanism for efficient segmentation. The core idea is dynamically sampling multi-resolution patches centered on objects, which is biologically inspired and aims to reduce computational cost while maintaining accuracy. The authors demonstrate good performance on multiple benchmarks, emphasizing parameter efficiency and scale invariance, particularly for small objects.

**Strengths:**

- The fovea-like patching mechanism is interesting, which addresses the inefficiencies of full-image encoding.

- The scale-invariant design is robustly validated on the proposed ObjaScale dataset, where FLIP outperforms SAM variants by large margins.

- The hierarchical inference scheme further enhances practicality for real-time applications.

**Weaknesses:**

- The reliance on a 2D Gaussian prior derived from ground-truth masks during training raises concerns about its generalizability to real-world scenarios, where object shapes are often irregular or annotations are imperfect. This dependency warrants further discussion.
- While FLIP demonstrates excellence in handling small objects, its performance on large-scale or complex occluded objects remains underexplored. The comparison is limited to SAMv1, omitting SAMv2 and recent object-centric models such as SlotAttention or diffusion-based segmenters. This limited comparison undermines the claim of broad superiority.
- Although the paper presents computational efficiency claims supported by inference times on an RTX 4090, a comprehensive analysis of memory footprint across diverse hardware is lacking.
- The synthetic ObjaScale dataset may not accurately capture real-world noise and variability. The paper fails to discuss potential domain shift issues when applying FLIP to unseen natural images. Moreover, the dataset evaluation lacks sufficient detail on construction methodology and scale distribution, making it challenging to assess the claimed advantages for tiny objects.
- The sparse training strategy focuses on boundary regions, but the adaptive sampling heuristic is not compared to alternative methods, and its impact on long-tail object categories remains unclear.
- Recent vision models, such as TransNeXt (CVPR'24) and OverLoCK (CVPR'25), have explored foveal mechanisms, but are not included in the related work. Additionally, DAT (CVPR'22) has proposed a deformable attention mechanism that can focus on a limited number of salient tokens, which is very similar to the mechanism proposed in this paper although this work focuses on patch embedding.
- The paper contains numerous typos and the citation format does not conform to the style of ICLR.

**Questions:**

Please address the weakness section.

---

> ### Author Response · Authors · 2025-11-17
> **Answer to Reviewer jYnH**
>
> We thank Reviewer jYnH for the constructive feedback. Below we provide a TLDR and also our point-by-point responses.
>
> ---
> ## TLDR:
> The 2D Gaussian is an unlearned location prior; FLIP generalizes to unseen real-world data including long-tailed object categories in LVIS, we now evaluate SAMv2, but Slot Attention models are out-of-scope baselines, and the missing related work is now cited.
>
> ---
>  - We acknowledge that the reliance on 2d Gaussian prompts is not as straight forward to adopt in a potential application and further research is needed to fully adopt FLIP in a potential object detection / tracking scenario. We now make this more clear in our conclusion section (See our latest revision).
>  - SAM V2 was not included in our evaluations (now it is), since it was trained on different data SA-1B and SA-V, while all models in our evaluation including FLIP were only trained on SA-1B. This means SAM V2 was trained on almost double the number of high quality human curated masks (10.2M in SA-1B and 10.0M in SA-V). It may also be noted that the STT model mentioned by two other reviewers is not compared to SAM V2. Nevertheless we now include SAM V2 in our evaluations and FLIP-Tiny still outperforms SAM2-H (see Table 1 in our latest revision).
>  - Slot Attention or derivative work has no means of prompting for an object segmentation mask thus was excluded from evaluations.
>  - We acknowledge that we did not do a memory footprint analysis across diverse hardware, but we also did not make any claims regarding memory footprints. The results of our reduced parameter count 0.5M compared to 641.1M stands on its own. Additionally, we think that our exemplary analysis on a single NVIDIA RTX-4090 is enough to support our speedup and accuracy claims.
>  - ObjaScale is deliberately designed as a synthetic toy dataset to test extreme scale invariance. FLIP was never trained on ObjaScale or on any of the other evaluation datasets.
>  - We specifically evaluate all models on a subset of objects of each dataset that are smaller than 1\% of the image area (see supplementary section in the paper).
>  - FLIP was never trained on any of the evaluation datasets. So every image in every evaluation set was never seen by FLIP during training. Thus we do show that FLIP is invariant to domain shift in unseen real world images.
>  - We evaluate FLIP on LVIS, which is a dataset specifically designed to test segmentation performance on long tail-object categories. Thus, we do show that FLIP achieves strong performance on long-tail object categories.
>  - We thank the reviewer for pointing out the other related work like TransNeXt and OverLoCK and DAT and have included them in the related work section (see our latest revision).
>  - We carefully proofread our paper again and also carefully checked that all our citations conform to the style of ICLR (see our latest revision).

---

> > ### Comment · Reviewer_jYnH · 2025-11-18
> >
> > Thank you for your response. Some of my concerns have been addressed.
> >
> > However, a major concern persists regarding the technical contribution of this work, which fails to clearly distinguish itself from previous works, including STT and FLIP. Furthermore, the authors merely mention the similar works I listed in the related works section without elaborating on how their approach differs or improves upon existing methods, particularly with regards to the deformable attention mechanism, which also focuses on a limited number of salient tokens. At the very least, theoretical analysis or empirical results should be provided to substantiate the claims.
> >
> > Moreover, I have noticed that other reviewers (XpqH and kR6o) have raised similar concerns, with which I fully agree.

---

> > > ### Author Response · Authors · 2025-11-26
> > > **Answer to Reviewer jYnH**
> > >
> > > Thank you for the follow-up. To clarify, the method under review is FLIP (Fovea-Like Input Patching).
> > >
> > > Table 1 in the revised manuscript reports the cross-dataset results, and the Appendix summarizes ablations of our architectural components.
> > >
> > > Algorithm 1 and the added figures formalize the foveated patch sampling so that readers can directly compare it to related deformable and foveated mechanisms.
> > >
> > > We also evaluate FLIP with 2D Gaussians derived from bounding boxes (see Table 1). FLIP-Tiny remains ahead of SAM2-L on average despite the weaker prompt signal.
> > >
> > > Following Reviewer kR6o's remark about an unfaithful prompt mismatch, we do not present a head-to-head STT comparison, instead, we discuss the differing task setups in the other responses.
> > >
> > > Please also refer to our general comment to all reviewers summarizing the updated performance gains and comparisons.

---

> > > > ### Comment · Reviewer_jYnH · 2025-11-26
> > > >
> > > > Thank you for your efforts in the rebuttal process. Sorry for the previous typo. I have carefully read the newly updated response, where the authors mention that "FLIP is the first model that only samples locally and object-centric mask encoding". However, I had listed several works in my original comments that have already explored similar foveal mechanisms. Therefore, this mechanism is not a novel contribution of this paper. Moreover, the authors have not adequately addressed the concerns I raised. Considering the common issues pointed out by other reviewers, I still believe that the overall contribution of this paper is somewhat incremental. Therefore, I have decided to maintain my original score.

---

> > > > > ### Author Response · Authors · 2025-11-26
> > > > > **Response to Incrementality & Novelty (Reviewer jYnH)**
> > > > >
> > > > > It appears that reviewer jYnH is highly motivated to prevent our paper from being published. We do not believe that the reasoning provided is warranted any longer.
> > > > >
> > > > > The argument of ‘incrementality’ could be applied to virtually any model. In the reviewer’s interpretation, the mere use of the term ‘fovea’—or, more generally, any mechanism that locally enhances contrast—seems to be taken as evidence of incrementality.
> > > > >
> > > > > Two further points that make the argument outright wrong (either one should be sufficient on its own):
> > > > >
> > > > > **Historical perspective**: Foveated vision models have existed since the 1980s (e.g., log-polar transforms in Schwartz, 1987; the neocognitron in Fukushima, 1980, has local enhancements; and many others throughout the computational neuroscience literature). In this sense, all of the papers cited by the reviewer, as well as our own work—and even very recent models such as STT—are incremental to some degree.
> > > > >
> > > > > **Contrast to the mentioned models**: The three papers referenced by reviewer jYnH indeed use the term ‘fovea’ or incorporate some means of locally enhancing contrast. However, none of them employs a mechanism that is similar to ours. Our FLIP model avoids processing all pixels through local patch sampling. This is precisely the point we made in our previous response. We already cite these three papers in our newest revision, but it remains the case that they do not implement the mechanism we introduce.
> > > > >
> > > > > In conclusion, the reviewer’s assertion of incrementality is simply incorrect—or, viewed from another perspective, is appliclable to any paper that is published nowadays.

---

### Official Review · Reviewer_kR6o · 2025-11-01

**Soundness:** 3
**Presentation:** 3
**Contribution:** 2
**Rating:** 4
**Confidence:** 4

**Summary:**

This study introduces a learnable fovea-like input patching (FLIP) scheme to focus on only important regions in an image before encoding. This design is scale-invariant and saves computation. FLIP is able to achieve 1000x fewer parameters while keeping the same accuracy with SAM.

**Strengths:**

- This paper is presented neatly with a clear motivation. It starts with a motivation from biological fovea structure and then introduces a learnable 2D Gaussian distribution as possible focal regions.
- The proposed fovea patching does not require encoding the full image, which is able to significantly accelerate inference and training. This is verified on different benchmarks as it achieves the same performance with SAM while using only 1000x fewer parameters and being 6x faster during inference.

**Weaknesses:**

- Novelty issues. The idea that using explicit samping patches at multiple resolutions and then embedding them with resolution-specific modules has been proposed in the literature on different tasks. The most relevant one to this study is STT[1], which proposes foveating the input as a way to tokenize images efficiently for point-prompted segmentation. However, this study is not cited nor referred to in the manuscript.
- The structural modifications as described in Sec 3.2 and 3.3 are mostly intuitive and incremental. The motivation of the introduction of independent LNs for Q, K, and V is not described in the manuscript, which leads to confusion to the reviewer. According to Fig. 3b, independent LNs are sequentially applied to X.  However, the features have already been normalized after the first LN, and the last two are thus equivalent to merely adding a new scaling and bias fator compared to the first one, which can actually be also learned in the weights of Q, K, and V. The necessity of adding independent LNs sequentially is thus under concern. Moreover, various modifications in regards to the normalization in transformers have been discussed in the literature, e.g.  [2], and should be correctly cited in the manuscript.
- The paper claims a 6x accelerance over SAM. However, the mere comparison with SAM is considered not comprehensive enough as SAM is designed on a much broader downstream application scales. Therefore, FLIP is recommended to validate itself over more datasets (e.g. the 23 datasets as in SAM). The generalization ability is a special concern to the reviewer as the Gaussian distribution parameters are learned from the training datasets and irrelevant patches are filtered out.

 [1] Schmidt, Tanner, and Richard Newcombe. "Segment This Thing: Foveated Tokenization for Efficient Point-Prompted Segmentation." Proceedings of the Computer Vision and Pattern Recognition Conference. 2025.
[2] Menary, Stephen, Samuel Kaski, and Andre Freitas. "Transformer Normalisation Layers and the Independence of Semantic Subspaces." arXiv preprint arXiv:2406.17837 (2024).

**Questions:**

The reviewer look forwards to the authors rebuttal in regards to its novelty issues and recommends the authors to compare the results with the network in SST as well.

---

> ### Author Response · Authors · 2025-11-17
> **Answer to Reviewer kR6o**
>
> We thank Reviewer kR6o for the constructive feedback. Below we provide a TLDR and also our point-by-point responses.
>
> ---
> ## TLDR:
> STT is concurrent work and fundamentally different (point-prompted grid tokenization in large models vs. FLIP's off-grid Gaussian-prompted sampling with model-size-driven speedups); we now cite and evaluate STT and clarified that Q/K/V LNs are parallel.
>
> ---
> ## Novelty issues:
> We where not aware of the STT paper, but have cited and evaluated it in our latest revision. We want to stress the point that STT and FLIP where developed concurrently, the arXiv timestep of STT is 10 Jun 2025.
>
>
> Apart from this, it needs to be pointed out that STT does outperform Sam/Efficient Sam ONLY on ONE of the nine reported datasets (two ties, six times SAM-H/SAM-L or EfficientSAM S win - partially with relatively large margins, cf. Table 5 in STT paper)!
> Our FLIP system outperforms SAM/Efficient SAM on ALL of the six reported datasets.
>
>
> Further, FLIP is by no means incremental when compared to STT.
> While FLIP and STT are both inspired by foveation, they approach the single object segmentation vastly different: STT starts with a point prompt and divides a cropped input grid centered on this point into increasingly bigger patches. The starting minimum patch size of STT is 16x16 and all bigger patches are down-sampled to this 16x16 patch size. In contrast our FLIP model extracts patches that don't lie on a grid but are sampled from a real valued 2d gaussian distribution which is our input prompt. The sampled patches in FLIP are then extracted from the image exactly at the sampled patch position using bilinear interpolation from the image pixel grid to the patch coordinates. Additionally, our maximum patch size is 16x16 and our smallest patch size is 1x1 in our ablations, we specifically show that using these smaller patch sizes drastically improves performance (see supplementary section of the paper).
>
>
> STT uses a much bigger transformer with 90.11M (STT-B), 307.67M (STT-L) and 635.34M (STT-H) parameters and their main speedup comes from using fewer tokens (around 23.8 times less than SAM), in contrast FLIPs main speedup comes from reduced parameter counts. Our smallest model FLIP-T with 0.5M parameters (more than 1000 times less than SAM-H) already outperforms SAM-H on average (in 3 out of 6 datasets, SAM-B in 5 out of 6 datasets with still nearly 200 times less parameters).
>
>
> A fair comparison of our FLIP approach with STT is not directly possible as STT only uses point prompts. Point prompts provide much less information about the target object than bounding box prompts or 2d Gaussian prompts and are in fact ambiguous (which is clearly stated in both the STT and SAM papers). This is a big disadvantage for STT when compared to FLIP. Nevertheless, we ran STT through our evaluation pipeline and adopted the same point prompt selection as used in the STT paper (selecting the point inside the mask that is the farthest away from any mask boundary). As expected STT performed much worse in comparison to the other models that use unambiguous bounding box or 2d Gaussian prompts.
>
> ---
> ## Layer Norms:
>  - The three LNs are parallel, not stacked. Each branch (Q/K/V) receives its own LN with separate $(\gamma$, $\beta$). We updated the figure in our revision to make this more clear.
>  - The three independent layer norms empirically improve the performance of our hierarchical inference method in comparison with a single LN for Q,K and V. Compare $LaPE-SNorm_h$ with the baseline in our ablations.
>  - We agree that the structural modifications are mostly incremental and intuitive. The novelty of our method does not stem from these transformer building blocks, but form our novel foveated object-centric input processing.
>  - We now also cite the paper "Transformer Normalisation Layers and the Independence of Semantic Subspace".
>
> ---
> ## Paper claims:
>  - Our paper targets the specific single object prompt application (like STT), not the whole range of SAM applications.
>  - We show that in the specific setting of segmenting a single object our model outperforms SAM in terms of accuracy with a 6x speedup.
>  - We extensively evaluated all models reported in the paper including all SAM variants ourselves on 5 standard and widely used object segmentation datasets and our own ObjaScale dataset. All model evaluations where performed on the same hardware a Nvidia RTX-4090 with identical settings. We think this is enough to support our claim of a 6x speed improvement. Evaluating on additional datasets will not significantly change the results.
>  - The 2d Gaussian is not learned but is used as an input prompt. At the level of sampling patches from the 2d Gaussian no "irrelevant" patches are filtered out. They are just sampled from the 2D Gaussian distribution. We only filter out strongly overlapping patches. The 2d Gaussian prompt just sets a location prior similarly to the STT papers point prompt.

---

> > ### Comment · Reviewer_kR6o · 2025-11-18
> >
> > I appreciate the authors' response.
> > ## Resolved concerns
> > - Appropriate reference of STT.
> > - Parallel layer norms of Q,K, and V (The original Figure 3 was indeed misleading).
> > - FLIP is light-weighted and achieves higher performance with STT at the same inference cost.
> >
> > ## Unresolved / New concerns
> > - The organization of the manuscript should be improved. The highlight of this study is foveal patching, and the structural improvement is minor (also acknowledged by the authors). However, the former comprises of only 50 lines in the method part (line 158-206) while the latter spans twice as much (line 208-318). This priority layout not only results in the missing important details in foveal patching but also distracts the readers. I would like to recommend adding more important details regarding to foveal patching in the main part and put some of the structural details into the supplementary materials.
> > - This study has two independent motivations: foveal patching & structural modifications. Even though the combination brings in higher performance, it is short of delivering a compact and coherent story. It is also hard for the reviewers to locate the true advantage of the Gaussian fovea design in this study. For example, as for the performance comparison between FLIP and STT, the authors stated in the rebuttal that "FLIP's main speedup comes from reduced parameter counts", yet what I expected was the benefits of the Gaussian foveal patching instead of the reduced parameters.
> > - The revised comparison between STT and FLIP in Table 1 is biased and may even appear desrespectful for STT as the GT mask is used to construct the Gaussian prompts in FLIP while only one point prompt is used in STT. We understand that STT only support point prompt and cannot be directly compared. I would suggest a best-effort fair comparison (e.g. STT uses the gt box as its crop) or no direct comparison at all, both at least better than the current version of Table 1.
> > - One small question. The inference speedup of STT is around 20-25 times compared to SAM (Table 2 in STT), while it is reported a only 15x speedup in Table 1 in this study, which could probably result in a lower baseline. I wonder what could be the reason apart from different GPU types.
> > - I also agree with Reviewer XpqH on the true application of the Gaussian prompts. The authors are recommended to justify the motivation and possible applications of Gaussian prompts. One personal experience in image annotation is that annotating a point for one object can be roughly 3-5 times faster than annotating a rectangular bbox, not to mention a Gaussian eclipse. Therefore, the potential impact also needs to be addressed carefully.

---

> > > ### Author Response · Authors · 2025-11-26
> > > **Answer to Reviewer kR6o**
> > >
> > > We thank the reviewer for suggesting to highlight and explain the foveal patching more clearly. We now moved the detailed structural component explanation to the appendix and improved the patch sampling subsection by adding a formal algorithmic description and two additional figures.
> > >
> > > The wording in our rebuttal was misleading: we did not mean that FLIP’s speedup is an independent consequence of having fewer parameters. Rather, the fovea-like local sampling is what enables a much smaller encoder, and the reduced parameter count is a direct consequence of this design.
> > >
> > > We acknowledge that the comparison between FLIP and STT may appear disrespectful since both methods cannot be compared on equal ground, and thus we opted to remove STT from Table 1 as suggested.
> > >
> > > STT excludes pre-processing time for all models as stated in their paper in Figure 2. In our evaluation pre-processing time is incorporated for all models, which could lead to the difference in inference speedup.
> > >
> > > We see the potential applications for Gaussian prompts not in human annotations but in potential future tracking applications. It is conceivable that a tracking model build an top of FLIP could predict 2d Gaussian as an estimation of the objects location and orientation in the next frame. This falls out almost naturally of a Kalman-filter-based tracker, as its prediction step already propagates a Gaussian belief over the object state, and projecting the predicted mean and covariance into image space directly yields the parameters of such a 2D Gaussian prompt.
> > >
> > > We acknowledge the difference in our 2d Gaussian prompting to a more straight forward bounding box prompting and have fine tuned our model to use Gaussian's derived from bounding box prompts, which results in slightly lower performance but FLIP-Tiny still beats SAM2-L on average with only 0.5M parameters. Additionally to mach compute we also fine tuned a separate set for all our models with the 2d gaussian prompts, which resulted in even higher performance than we previously reported. (see also our general comment to all reviewers at the beginning of the rebuttal)

---

### Author Response · Authors · 2025-11-26
**Official Comment to All Reviewers**

Dear reviewers of our Looking Locally paper (FLIP model).

We have taken your comments very seriously and realized that parts of the criticism was warranted:
we have compared FLIP, which samples image patches from Gaussians derived from the ground-truth mask, with the other models, which receive the exact bounding box.
This was not a fully fair comparison.

Meanwhile, your criticism also made us realize that we had not fine-tuned our model to the test situation. Until now, we had varied - for data augmentation purposes - the amount of patches and the sampling density during training.

Over the last 10 days, we have thus fine-tuned our FLIP variants to optimize its performance when being confronted with bounding box-like information (converted by fixed factors into mean and standard deviations of an isotropic Gaussian).
Moreover, during this fine tuning, we have removed the data augmentation part.
Still, we trained and fine-tuned on SAM's SA-1B dataset only (as before).

The updated results in Table 1 show an improvement of the original FLIP model - partially scoring nearly 2 points higher now - outperforming all SAM variants even more on all six datasets.
$FLIP_{bb}$ - the version that is optimized for Isotropic Gaussian prompts given the bounding box information only - scores slightly lower than the results we gained with the previous non-fine-tuned FLIP version.
Nonetheless, $FLIP_{bb}$ still outperforms all SAM variants in two out of the six datasets and draws nearly even on the other four - on average still being clearly superior to all SAM variants.

We want to emphasizes that performance (and smaller size and shorter computational time) is not everything.
FLIP is the first model that only samples locally from an image.
It separates location from appearance, yielding local, object-centric mask encodings.
The model design is novel and unique - and even inspired from computational cognitive neuroscience (for some readers this may be of more while for others of less interest).

Accordingly, in the main text we now focus more on introducing the novel Fovea-like input patching mechanism and other core processing components, which induce an object-centric inductive bias, so that specifying an object’s position leads to an object-centered latent encoding. We hope this clarifies the uniqueness of our approach.

Thank you again for your consideration and careful reviews. We hope that you agree that pushing the evaluations and comparisons even further confirms the robustness and future potential of our approach.

We thus hope that the reviewers are now convinced that this paper should indeed be published at ICLR.

---

### Meta-Review · Area_Chair_g1Wu · 2025-12-24

**Summary:**

This work proposes fovea-like, prompt-guided local patch sampling for efficient object-centric segmentation, claiming large parameter and speed gains with competitive accuracy. Reviewers’ major concerns were about novelty vs. closely related foveated/tokenization work (esp. STT and related mechanisms), fairness/practicality of the prompting setup (Gaussian prompts initially derived from GT masks vs box/point prompts for baselines), and insufficiently clear/formal description of the core patch sampling in the initial version. One reviewer was strongly positive on the empirical efficiency/accuracy tradeoff and the ObjaScale benchmark; others remained borderline negative.
Despite clear empirical efficiency gains, the submission’s core contribution remains insufficiently distinguished from closely related foveated/tokenization and deformable-attention lines of work, and the central Gaussian-prompted evaluation leaves lingering concerns about practical prompting realism and comparison fairness.

**Reviewer Concerns:**

Addressed or Partially Addressed:
1. Explained and added positioning vs. STT and related work.
2. Clarified Q/K/V LayerNorms are parallel and not stacked.
3. Improved description of foveated patching and reorganized some method details.
4. Added evaluation with box-derived Gaussians to reduce prompt-mismatch concerns, and also incorporated SAMv2 results.

Unresolved:
1. Novelty concerns persist for multiple reviewers (especially jYnH and XpqH): they view the core idea as overlapping prior foveated/sparse mechanisms, with differences mostly architectural/implementation.
2. Realistic prompting remains unclear: even with box-derived Gaussians and a tracking argument, the practical pipeline and sensitivity to noisy prompts are not fully nailed down.
3. Comparisons to other recent efficient segmentation baselines beyond the SAM family remain limited.

**Reviewer Scores:**

Scores likely remain largely unchanged. Reviewer FDjP was already strongly positive, while the other three reviewers (kR6o, jYnH, XpqH) are likely to maintain borderline-negative assessments, mainly due to concerns about novelty, fairness of the prompting setup, and practical applicability. Although the rebuttal clarified several technical points and improved the presentation, it did not appear to successfully change their overall judgments.

---

### Decision · Program_Chairs · 2026-01-26

Reject